



# Authigenic formation of Ca-Mg carbonates in the shallow alkaline Lake Neusiedl, Austria

Dario Fussmann[1], Avril Jean Elisabeth von Hoyningen-Huene[2], Andreas Reimer[1], Dominik Schneider[2], Hana Babková[3], Robert Peticzka[4], Andreas Maier[4], Gernot Arp[1], Rolf Daniel[2] and Patrick Meister[3]

[1]Geobiology, Geoscience Centre, Goldschmidtstraße 3, Georg-August-University Göttingen, 37077 Göttingen, Germany
[2]Genomic and Applied Microbiology Genomics Laboratory, Institute of Microbiology and Genetics, Grisebachstraße 8, Georg-August-University Göttingen, Göttingen, 37077 Germany
[3]Department of Geodynamics and Sedimentology, University of Vienna, Althanstraße 14, 1090 Vienna, Austria
[4]Department of Geography and Regional Research, University of Vienna, Althanstraße 14, 1090 Vienna, Austria

*Correspondence to*: Dario Fussmann (dario.fussmann@uni-goettingen.de)

**Abstract.** Despite advances regarding the microbial and organic-molecular impact on nucleation, the formation of dolomite in sedimentary environments is still incompletely understood. Since 1960, apparent dolomite formation has been reported from mud sediments of the shallow, oligohaline and alkaline Lake Neusiedl, Austria. To trace potential dolomite formation or diagenetic alteration processes in its deposits, lake water samples and sediment cores were analyzed with respect to sediment composition, hydrochemistry and bacterial community composition. Sediments comprise 20 cm of homogenous mud with 60 wt% carbonate, which overlie dark-laminated consolidated mud containing 50 wt% carbonate and plant debris. Hydrochemical measurements reveal a shift from oxic lake water with pH 9.0 to anoxic sediment pore water with pH 7.5. A decrease in $SO_4^{2-}$ with a concomitant increase of $\Sigma H_2S$ and $NH_4^+$ from 0-15 cm core depth, indicates anaerobic heterotrophic decomposition, including sulfate reduction. The bacterial community composition reflects the zonation indicated by the pore water chemistry, with a distinct increase of fermentative taxa below 15 cm core depth.

The water column is highly supersaturated with respect to (disordered) dolomite and calcite, whereas saturation indices of both minerals rapidly approach zero in the sediment. Notably, the relative proportions of different authigenic carbonate phases and their stoichiometric compositions remain constant with increasing core depth. Hence, evidence for Ca-Mg carbonate formation or ripening to dolomite is lacking within the sediment of Lake Neusiedl. As a consequence, precipitation of high-magnesium-calcite (HMC) and very-high-magnesium-calcite (VHMC) does not occur in association with anoxic sediment and sulfate reducing conditions. Instead, analytical data for Lake Neusiedl suggest that authigenic HMC and VHMC precipitate from the supersaturated, well-mixed aerobic water column. This observation supports an alternative concept to dolomite formation in anoxic sediments, comprising Ca-Mg carbonate precipitation in the water column under aerobic and alkaline conditions.



## 1. Introduction

Dolomite (CaMg[CO$_3$]$_2$) is the most abundant carbonate mineral in Earth's sedimentary record. It has rarely been observed forming in recent environments. Instead, most occurrences of large dolomite deposits in the geological record are the result of pervasive dolomitization of precursor carbonates by fluids with high Mg:Ca ratios and temperatures during burial (e.g. Machel, 2004). In contrast, the formation of dolomite near the sediment surface, so-called penecontemporaneous dolomite (Machel 2004 and references therein), or even primary precipitation in shallow aquatic environments, are often difficult to trace in the rock record and capture in modern environments. The difficulty in capturing ongoing dolomite formation is due to its peculiar kinetics, which are still incompletely understood, despite intense laboratory and field experiments. Dolomite does not form in sites where sufficient Ca, Mg, and carbonate ions are provided, which is generally explained by the high kinetic barrier of dolomite nucleation and growth (e.g. Lippmann, 1973).

Based on the presence of sulfate-reducing bacteria, Vasconcelos et al. (1995) proposed a microbial model, in which sulfate-reducing bacteria mediate carbonate precipitation, while Brady et al. (1996) consider sulfate ions as inhibitors for dolomite growth. Further experiments were performed with various different organisms, such as denitrifiers (Rivadeneyra et al., 2000), methanogenic archaea (Roberts et al., 2004) and aerobic halophilic bacteria (Sánchez-Román et al., 2009). All of these studies showed aggregate formation of carbonate minerals with the characteristic $d_{104}$-peak of dolomite under X-ray diffraction, hence, supporting a microbial factor in dolomite formation. It has been hypothesized that dolomite nucleation is mediated by microbial extracellular polymeric substances (EPS; Bontognali et al., 2014). However, Gregg et al. (2015) re-analyzed the X-ray diffraction data of many of the aforementioned microbial experiments, demonstrating that microbial dolomite products lack typical ordering reflections in XRD spectra and are in fact very high-magnesium-calcite (VHMC or "protodolomite"). In further studies sulfide (Zhang et al., 2013b) dissolved organic matter (Frisia et al., 2018) or clay minerals (Liu et al., 2019) were suggested to favor protodolomite nucleation in porefluids. Nevertheless, it is not entirely clear, which of these factors play a fundamental role in natural environments and how the specific reaction mechanisms work.

While the concept that dolomite forms within sediments mediated by anaerobic microbial processes and their extracellular polymeric substances, is widely acknowledged, another aspect should be taken into account: The site of dolomite formation may not always coincide with the location where the mineral is found due to relocation after precipitation. Several studies describe unlithified dolomite precipitation in warm, arid and hypersaline marine environments, like coastal sabkhas (Illing et al., 1965; Bontognali et al., 2010; Court et al., 2017), coastal lakes, such as Lagoa Vermelha in Brazil (Vasconcelos and McKenzie, 1997; van Lith et al., 2002; Sánchez-Román et al., 2009; Bahniuk et al., 2015) and ephemeral lakes along the Coorong Lagoon in South Australia (von der Borch, 1976; Rosen et al., 1989; Warren, 1990; Wright and Wacey, 2005). Dolomite precipitation is further reported in endorheic hypersaline lakes, e. g. Lake Qinghai in Tibet (Deng et al., 2010), Lake Acigöl (Turkey; Balci et al., 2016) and alkaline playa lakes such as Deep Springs Lake in California (Meister et al., 2011).

Another Ca-Mg carbonate forming location can be found in Turkey, where McCormack et al. (2018) describe dolomite in Quaternary sediments from Lake Van, which is suggested to have formed at the sediment-water interface including varying salinities and low temperatures. These dolomite-bearing deposits have been related to the onset of a falling paleo lake-level and, hence, changing hydro-chemical conditions. Importantly, McCormack et al. (2018) locate the formation of dolomite near the sediment-water interface, where it is presumably related to microbial EPS. However, this area is also exposed to significant fluctuations in pH, temperature, and





supersaturation. According to precipitation-experiments conducted by Deelman (1999), dolomite can form due to
such fluctuations in pH and temperature, thereby breaking Ostwald's step rule via undersaturation of other
metastable carbonate phases.
Lake Neusiedl is a Ca-Mg carbonate precipitating water body with exceptionally low salinity (1-2 g L$^{-1}$). It is a
shallow and seasonally evaporative lake in the proximity of Vienna, Austria. Schroll and Wieden (1960) first
reported the occurrence of poorly crystallized dolomite (notable by its broad XRD-reflections) at this locality and
Müller et al. (1972) related its formation to diagenetic alteration of high-magnesium-calcite (HMC). The Mg:Ca
ratios in Lake Neusiedl are unusually high (>7) compared to freshwater lakes, which favor the precipitation of
HMC (Müller et al., 1972). Little is known about the crystallization paths of the Ca-Mg carbonate phases in this
lake, in particular whether they form in the anoxic sediment or oxic water column and if early diagenetic alteration
to dolomite ("ripening") takes place.
We revisit the formation of dolomite in Lake Neusiedl by comparing the sediment-geochemical and *in-situ* pore
water data and critically evaluating the location of precipitation. This approach has been used to study dolomite
formation in Lagoa Vermelha (van Lith et al. 2002; Moreira et al., 2004) or in Deep Springs Lake (Meister et al.,
2011). Since 2005, *in-situ* pore water extraction via rhizon samplers has been applied for geoscientific research
questions (Seeberg-Elverfeldt et al., 2005) and several in situ pore water studies were conducted using this
technique (e.g. Bontognali, 2010; Birgel et al., 2015; Steiner et al., 2018). A comparable *in-situ* pore water data
set for an oligohaline seasonally evaporative lake, which addresses the question of authigenic Ca-Mg carbonate
precipitation, is absent so far. We further provide bacterial community analyses to address the potential role of
microbes and their metabolisms in a carbonate mineral precipitation or alteration pathway. Hence, our study has
three goals: (i) finding indications for the origin of Ca-Mg carbonate formation, (ii) evaluating the microbiological
and geochemical conditions and their influence on carbonate saturation, and (iii) discussing which factors drive
the formation of Ca-Mg carbonates in Lake Neusiedl.
**2. Study Area**
Lake Neusiedl, situated at the Austrian-Hungarian border, is the largest endorheic lake in Western Europe. It is
located in the Little Hungarian Plain, a transition zone between the Eastern Alps and the Pannonian Basin in central
Hungary. The region has been tectonically active since the early Miocene (Horváth, 1993) and is affected by NE-
SW trending normal faults. This early Miocene tectonic activity included the closing of the Central Paratethys Sea
and the formation of Lake Pannon about 11.6 million years ago. This ancient water body was characterized by
highly fluctuating water levels that caused the deposition of local evaporite layers, which influence the salinity of
today's deeper aquifers in the area (Piller et al., 2007; Krachler et al., 2018).The present topography of the Little
Hungarian Plain is the result of ongoing local uplift and subsidence, which commenced in the latest Pliocene
(Zámolyi et al., 2017). Elevated regions are represented by the Rust- and Leitha Hills, which are horst-like
structures located west of Lake Neusiedl. Northward, the water body is separated from the Vienna basin by the
raised Parndorf Plateau, which has a 25-45 m higher surface elevation than the lake area. South- and eastward,
Lake Neusiedl is surrounded by flats, namely the Hansag- and Seewinkel Plain. Despite its proximity to the Alps,
the region surrounding Lake Neusiedl did not have an ice cover during the last glacial maximum. Hence, its
morphology is shaped by periglacial erosion and sedimentation (van Husen, 2004). Throughout the Seewinkel
Plain, Pannonian marine to brackish sediments are largely covered by fluvioglacial gravels. The gravels thin out





westwards and are thus missing beneath parts of Lake Neusiedl, where fine-grained, unlithified lacustrine mud
directly overlies compacted Pannonian strata. The absence of a gravel layer has made the former lake area
vulnerable to aeolian erosion, favoring the formation of the present day flat trough over tectonic subsidence
(Zámolyi et al., 2017).
The surface area of the water body spreads over 315 km² with a maximum depth of 1.8 m. With a salinity of 1-2
$g \cdot L^{-1}$ and elevated pH values (>8.5), the water chemistry differs significantly from that of freshwater lakes
(salinity: < 0.5 $g \cdot L^{-1}$, pH: 6.5-7.5). Increased amounts of sodium- and bicarbonate ions mainly contribute to the
lake's soda-like character (Herzig, 2014). Furthermore, the Mg:Ca ratio is unusually high in comparisson to
freshwater lakes (Krachler et al., 2012). Permanent surface water inflow is mainly provided by the Rákos and the
Wulka streams, which drain a catchment area that is approximately 2.6 times the size of Lake Neusiedl (1,120
km²). Thus, their contribution to the lake's water balance is negligible compared to the significantly higher input
from precipitation, providing 80 – 90% of the lake water (Herzig and Dokulil, 2001). As a result of its shallowness
and the endorheic drainage system, the lake is very vulnerable to climatic changes, which highly influence the
water level, water volume and, hence, the surface area of the lake throughout the year and over the centuries. In
the past, Lake Neusiedl was characterized by highly fluctuating water levels and desiccation events (Moser, 1866),
the last of which dates back between 1865 and 1870. Since 1910, the lake's water outflow can be regulated by the
artificial Hanság- or Einser-Kanal in case of severe flooding events. The canal is located at the lake's southeastern
shore (Fig. 1).
More than half (178 km²) of Lake Neusiedl´s surface area is covered with reed. Due to its wind exposure and
shallowness, the water column of the open water area is well mixed and contains high amounts of suspended
particles. The wind sheltering effect of *Phragmite* spears, in contrast, leads to clearer water in the reed belt. Clastic
input into the water body is minor and reflects the mineralogical composition of the western neighboring Rust-
and Leitha hills, which are characterized by crystalline rocks of the Eastern Alpine basement and Miocene marine
carbonates ("Leithakalk", Fig. 1). The deposits forming the present bed of Lake Neusiedl consist of fine-grained
mud, which mainly contains typical authigenic carbonate phases such as Mg-calcite and dolomite/VHMC (Löffler,
1979). Those phases can clearly be distinguished from pure calcite, which is considered as allochthonous in the
sedimentary environment of Lake Neusiedl (Müller et al., 1972). It is noteworthy, that the mud volume has doubled
in the time from 1963 to 1988, leading to an increase of the volumetric mud/water ratio from 36:64 in 1963, to
49:51 in 1988. This mud layer covers the whole lake area and would yield an average thickness of 64 cm, assuming
an equal distribution across the lake basin (Bácsatyai, 1997). The soft sediment thickness can increase up to 1 m
at the border of the reed belt and open water, where *Phragmite* spears act as sediment traps for current driven,
suspended particles (Löffler 1979).

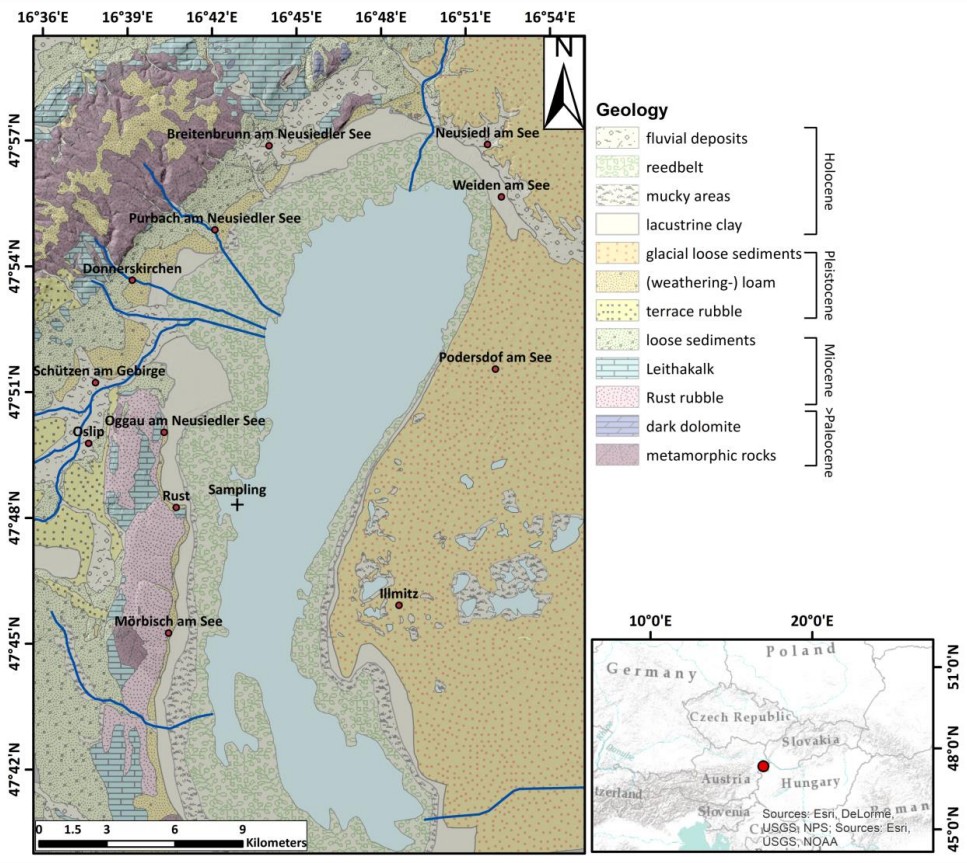

**Figure 1: Lake Neusiedl and its surrounding geology, redrawn and simplified after Herrman et al. (1993).**

## 3. Material and Methods

### 3.1 Sampling and field measurements

The sampling campaign at Lake Neusiedl was performed in August 2017 in the bay of Rust (16°42'33.635''E, 47°48'12.929''N) situated at the lake´s central western shore. A pedalo boat was utilized to enable sampling approximately 500 m offshore. Physicochemical parameters of the lake water were measured directly in the field using a WTW Multi 3430 device equipped with a WTW Tetracon 925 conductivity probe, a WTW FDO 925 probe for dissolved $O_2$, and a WTW Sentix 940 electrode for temperature and pH (Xylem, Rye Brook, NY, USA), calibrated against standard pH-buffers 7.010 and 10.010 (HI6007 and HI6010, Hanna Instruments, Woonsocket, RI, USA; standard deviation ≤ 2%). Lake water was retrieved from a depth of 10 cm with a 500 ml Schott-Duran glass bottle without headspace from which subsamples for anion, nutrient and total alkalinity determination were distributed into 100 mL polyethylene (PE) and 250 mL Schott-Duran glass bottles (Schott, Mainz, Germany), respectively. For cation analysis, a 50 mL aliquot was filtered through membrane filters with a pore size of 0.7 μm (Merck, Darmstadt, Germany) into a PE-bottle and acidified with 100 μl $HNO_3$ (sub-boiled). Total alkalinity was determined via titration within 3 hours after sampling using a hand-held titration device and 1.6 N $H_2SO_4$ cartridges (Hach Lange, Düsseldorf, Germany; standard deviation ≤ 1.5%).



Five sediment cores, with the sample codes LN-K01, LN-K02, LN-K03, LN-K04 and LN-K05, were retrieved
using PVC-tubes (6.3 cm diameter; Uwitec, Mondsee, Austria) in approximately 30 cm lateral distance. All cores
were 30 to 40 cm in length and were used for sediment, pore water and bacterial community profiling. Cores LN-
K01 and LN-K02 were subsampled and treated for bacterial community profiling as described in von Hoyningen-
Huene et al. (2019) directly after recovery. Cores LN-K03, LN-K04 and LN-K05 were hermetically sealed after
recovery and stored upright at temperatures close to their natural environment ($22 \pm 2$ °C). Effects of pressure
differences are neglectable in the present case, because the cores were sampled just below the lake floor.
**3.2 Petrographic, mineralogical, and geochemical analyses**
Two cores, labelled LN-K04 and LN-K05, were used for sediment geochemical and petrographic analyses.
Sediment dry density and porosity were calculated from the corresponding sediment weights and volumes. For
bulk organic and inorganic carbon content detection, sediment increments of 2.5 cm were subsampled from core
LN-K04. They were freeze dried and powdered with a ball mill, before they were measured by a LECO RC612
(Leco, St Joseph, MI, USA) multi-phase carbon and water determination device. For calibration, Leco synthetic
carbon (1 and 4.98 carbon%) and Leco calcium carbonate (12 carbon%) standards were used. The same increments
were utilized for CNS elemental detection, which was operated with a Euro EA 3000 Elemental Analyzer
(Hekatech, Wegberg, Germany). 2.5-Bis (5-tert-benzoxazol-2-yl) thiophene BBOT and atropine sulfate
monohydrate (IVA Analysetechnik, Meerbusch, Germany) were provided as reference material. Analytical
accuracy of all analyses was better than 3.3%.
XRD-analyses were conducted with identical increments at the Department of Geodynamics and Sedimentology
in Vienna by a PANanalytical (Almelo, Netherlands) Xpert Pro device (CuK$\alpha$ radiation, $2\theta$ refraction range of 2-
70°, and a step size of 0.01°). Semi-quantitative phase composition analysis was performed with Rietveld
refinement of peak intensities by using MAUD (version 2.8; Lutterotti et al., 2007). To ensure a better
reproducibility of the semi-quantitative XRD-analysis, Rietveld refined results were compared and correlated with
carbon data retrieved from the aforementioned LECO RC612 device.
In core LN-K05, sediment increments of 5 cm were subsampled for thin sectioning and light microscopic
observations. To ensure a continuous section, rectangular steel meshes, 5 cm in length, were placed along the
sediment column. These steel meshes, filled with soft sediment, were then embedded in LR White resin (London
Resin Company, Reading, United Kingdom), after a dehydration procedure with ethanol. During dehydration, the
sediments were treated with Sytox Green nucleic acid stain (Invitrogen, Carlsbad, CA, USA) to stain eukaryotic
cell nuclei and prokaryotic cells for fluorescence microscopy. Samples were cured for 24 hours at 60°C before
thin section preparation. The thin sections were ground down to a thickness of 40 to 50 µm and then capped with
a glass cover. Petrographic observations were conducted with a petrographic and a laser-scanning microscope
(Zeiss, Oberkochen, Germany, lsm excitation: 543 nm, 488 nm, 633 nm, laser unit: Argon/2, HeNe543, HeNe633).
For scanning electron microscopy, non-capped unpolished thin section fragments and freeze-dried loose sediment
from cores LN-K05 and LN-K04 were placed on 12.5 mm plano carriers and sputtered with a platinum-palladium
mixture. Field emission scanning electron microscopy was conducted with a Gemini Leo 1530 device (Zeiss,
Oberkochen, Germany) with a coupled INCA x-act (Oxford Instruments, Abingdon, United Kingdom) EDX
detector.

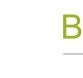
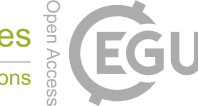

**3.3 Pore water analysis**

Redox potential and pH gradients were directly measured in the sediment of core LN-K03 one week after sampling with a portable WTW 340i pH meter, equipped with an Inlab Solids Pro pH-electrode (Mettler Toledo, Columbus, OH, USA) and a Pt 5900 A redox electrode (SI Analytics, Mainz, Germany) through boreholes (standard deviation $\leq 2\%$). Pore water was extracted from the core, using 5 cm CSS Rhizon samplers (Rhizosphere, Wageningen, Netherlands). Immediately after extraction, aliquots were fixed with Zn-acetate for determination of total sulfide ($\Sigma H_2S$). Pore water alkalinity was determined using a modified Hach titration method with self-prepared 0.01 N HCl cartridges as titrant. Major cation ($Ca^{2+}$, $Mg^{2+}$, $Na^+$, $K^+$ and $Li^+$) and anion ($Cl^-$, $F^-$, $Br^-$, $SO_4^{2-}$ and $NO_3^-$) concentrations of lake and pore water samples (including supernatants in the cores) were analyzed by ion chromatography with non-suppressed and suppressed conductivity detection, respectively (Metrohm 820 IC / Metrosep C3-250 analytical column, Metrohm 883 Basic IC/ Metrohm ASupp5-250 analytical column, Metrohm, Herisau, Switzerland; standard deviation $\leq 2\%$). Inductively coupled plasma mass spectrometry (ICP-MS; ICAP-Q, Thermo Fisher, Waltham, MA, USA) was used to determine Sr, Ba, Fe, Mn, Rb and B, as control for the cation determination by ion chromatography (standard deviation $\leq 3\%$).

Concentrations of $NH_4^+$, $NO_2^-$, $PO_4^{3-}$, $\Sigma H_2S$ and dissolved silica ($SiO_{2(aq)}$) were measured by photometric methods according to Grasshoff et al. (2009), using a SI Analytics Uviline 9400 spectrophotometer. In addition, methane and dissolved inorganic carbon (DIC) amounts were retrieved from a different core, sampled at the same locality in August 2017. Methane concentrations were determined from 5 cm$^3$ sediment samples stored upside down in gas-tight glass bottles containing 5 mL NaOH (5% w/v). Aliquots of 5 ml headspace methane were transferred to evacuated 10 ml vials. The aliquots were analyzed with an automated headspace gas chromatograph (GC Agilent 7697A coupled to an Agilent 7890B auto sampler) at the University of Vienna. Methane concentrations were quantified at a runtime of 1.798 min by a flame ionization detector and a methanizer. For linear calibration, a standard series with the concentrations 1001 ppb, 3013 ppb and 10003 ppb was used. DIC concentrations were retrieved by using a Shimadzu TOC-LCPH (Shimadzu, Kyoto, Japan) analyzer with an ASI-L autosampler and a reaction vessel containing a reaction solution of phosphoric acid ($H_3PO_4$, 25%). The DIC was measured by conversion to carbon dioxide, which was detected by a NDIR detector.

All measured values were processed with the PHREEQC software package (version 3; Parkhurst and Appelo, 2013). The implemented phreeqc.dat and wateqf4.dat databases were used in order to calculate ion activities and $pCO_2$ (partial pressure of $CO_2$) of the water samples and mineral saturation states. The saturation indices of mineral phases are given as $SI = \log (IAP/K_{SO})$.

**3.4 Bacterial 16S rRNA gene community profiling**

Two sediment cores labelled LN-K01 and LN-K02 were sampled for bacterial 16S rRNA gene-based community profiling. Each core was sampled in triplicate at every 2.5-5 cm of depth and the surface water filtered through a 2.7 (Merck, Darmstadt, Germany) and 0.2 µm (Sartorius, Göttingen, Germany) filter sandwich. RNAprotect Bacteria Reagent (Qiagen, Hilden, Germany) was immediately added to all samples, in order to preserve the nucleic acids. Before storage at -80°C, the samples were centrifuged for 15 min at 3.220 x $g$ and the RNAprotect Bacteria Reagent was decanted.

DNA was extracted and 16S rRNA genes were amplified and sequenced as described in detail by von Hoyningen-Huene et al. (2019). Briefly, DNA was extracted from 250 mg of each homogenized sediment sample or one third of each filter with the MoBio PowerSoil DNA isolation kit (MoBio, Carlsbad, CA, USA) according to

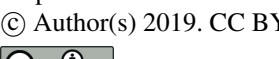



manufacturer's instructions with an adjusted cell disruption step. Bacterial 16S rRNA genes were amplified in
triplicate by PCR with the forward primer D-Bact-0341-b-S-17 and the reverse primer S-D-Bact-0785-a-A-21
(Klindworth et al., 2013) targeting the V3-V4 hypervariable regions. Primers included adapters for sequencing on
an Illumina MiSeq platform. PCR triplicates were pooled equimolar and purified with MagSi-NGS$^{Prep}$ magnetic
beads (Steinbrenner, Wiesenbach, Germany) as recommended by the manufacturer and eluted in 30 µl elution
buffer EB (Qiagen, Hilden, Germany).
PCR products were sequenced with the v3 Reagent kit on an Illumina MiSeq platform (San Diego, CA, USA) as
described by Schneider et al. (2017). Sequencing yielded a total of 6,044,032 paired-end reads, which were quality-
filtered (fastp, version 0.19.4; Chen et al., 2018), merged (PEAR, version 0.9.11; Zhang et al., 2013a) and
processed. This comprised primer clipping (cutadapt, version 1.18; Martin, 2011), size-filtering, dereplication,
denoising and chimera removal (VSEARCH, v2.9.1; Rognes et al., 2016). Taxonomy was assigned to the resulting
amplicon sequence variants (ASVs; Callahan et al., 2017) via BLAST 2.7.1+ against the SILVA SSU 132 NR
(Quast et al., 2012). After taxonomic assignment, 2,263,813 merged reads remained in the dataset. The resulting
ASV abundance table was used for the visualization of community gradients along the cores (von Hoyningen-
Huene et al., 2019). Data were analyzed using R (version 3.5.2; R Core Team, 2018) and RStudio (version 1.1.463;
RStudio; R Team, 2016) using the base packages. Extrinsic domains, archaea and eukaryotes were removed from
the ASV table for analysis. All ASVs with lower identity than 95% to database entries were assigned as
unclassified. Replicates for each depth were merged, transformed into relative abundances and all ASVs with an
abundance > 0.5% were summarized by their phylogenetic orders. Putative functions of all orders were assigned
according to literature on cultured bacterial taxa and the closest cultured relatives of the ASVs present in our
samples. For uncultured taxa, functions were inferred from literature on genomic and metagenomic sequencing
data (Suppl. Material Tab. S6). The resulting table with relative abundances and functional assignments was used
to generate bar charts in SigmaPlot (version 11; Systat Software, 2008).

## 4. Results

### 4.1 Sediment petrography and mineralogy

The cored sediment can be divided into three different lithological units. Unit I, in the first 15 cm below surface
(b.s.), is characterized by homogenous, light to medium grey mud with very high water content and porosity (>65
weight%, 0,67). The mud consists of very fine-grained carbonate and siliciclastics, largely in the clay and silt size
fraction. In the thin sections of embedded mud samples, carbonates make up most of the fine-grained matrix (Fig.
2A and B). Remnants of diatoms and ostracods occur with random orientation. Detrital grains up to fine sand
fraction, consisting of quartz, feldspar, mica, chlorite and carbonates make up as much as 20% of the sediment.
The latter are distinguishable from authigenic carbonate phases by their bigger (up to mm measuring) size and
fractured shape. The $C_{org}:N_{tot}$ ratio scatters around 10 (Fig. 3) and plant detritus is evident in thin sections as
opaque, up to several hundred µm in size, often elongated and randomly orientated particles (Fig. 2A and B).
These can be identified in the laser scan images, due to their chlorophyll related bright fluorescence (Fig. 4A and
B).
Unit II is located between 15 and 22 cm b.s. and appears as slightly darker, grey-colored mud without
macrostructures. The microcrystalline matrix appearance is similar to Unit I, however, phytoclasts and detrital
mineral grains are more abundant and up to mm in size, whereas the amount of bioclasts remains the same.





Noticeably, detrital carbonate minerals and quartz grains occur layer-like or in defined lenses (Fig. 2C and D). The
component to matrix ratio slightly increases up to 25:75% and cubic, small (up to 10 µm), opaque minerals often
occur intercalated with plant detritus. The $C_{org}:N_{tot}$ ratio also changes from 10 at 15 cm to 12 at 22 cm b.s..
Unit III, occurs from 22 to 40 cm b.s.. It is distinctly darker than the units above and shows a significant decrease
in water content and porosity to <50 weight% and <0.6, respectively. This decrease in porosity is also recognizable
by a more cohesive sediment texture. Lamination is visible at the core's outer surface, but not in the cut section, in
which plant detritus noticeably increases. Thin sections of this unit illustrate a rather compacted matrix, a
horizontal orientation of elongated phytoclasts and a layered structure with detrital mineral grains (Fig. 2E and F),
further supported by the laser scan image (Fig. 4C). Ostracod or diatom fragments still occur but are less abundant
than in the units above. The particle to matrix ratio increases up to 35:65% and the $C_{org}:N_{tot}$ ratio steadily increases
from 12 to 14 through Unit III.
In SEM images, the matrix appears as microcrystalline aggregate of several nanometer-sized clotted crumbs (Fig.
5). Locally, small, up to 1 µm in scale, irregularly shaped rhombohedral crystals are observable. With EDX
measurements, these tiny crystals were identified as Ca-Mg carbonate phases.
According to the XRD spectra, the bulk sediment mainly consists of carbonates and quartz with minor
contributions of feldspar, clay, and mica (Fig. 6). The $d_{104}$ peak shift provides a suitable approach to estimate the
Mg:Ca ratio in magnesium calcite and dolomite (Lumsden, 1979). Based on the $d_{104}$ peak positions, three carbonate
phases with different $MgCO_3$-content are present: A calcite phase with minor amounts of $MgCO_3$, a high-
magnesium-calcite phase (HMC) with circa 18 mole% $MgCO_3$ and a very-high-magnesium-calcite phase (VHMC,
Fig. 6). The latter shows a 104 peak, shifted from 31°2θ in ordered dolomite to ca. 30.8°2θ, indicating a $MgCO_3$
content of approx. 45 mole%. Estimated relative mineral abundances vary between the three units (Fig. 7): In Unit
I the amount of authigenic carbonate minerals remains relatively constant at 55 weight%, whereas in Unit II a
steep/large increase of detrital mineral phases (feldspar, quartz, calcite, mica) can be found. In Unit III the amount
of Ca-Mg carbonate phases decreases and scatters around 40 weight%. Mica slightly increase with depth below
23 cm. Nevertheless, the authigenic HMC to VHMC ratio does not change significantly throughout the section.
Notably, all authigenic Ca-Mg carbonate phases do not show any down-core trend in stoichiometry. The
Mg/(Ca+Mg) ratios of distinct solid phases remain largely constant with depth (Fig. 8).

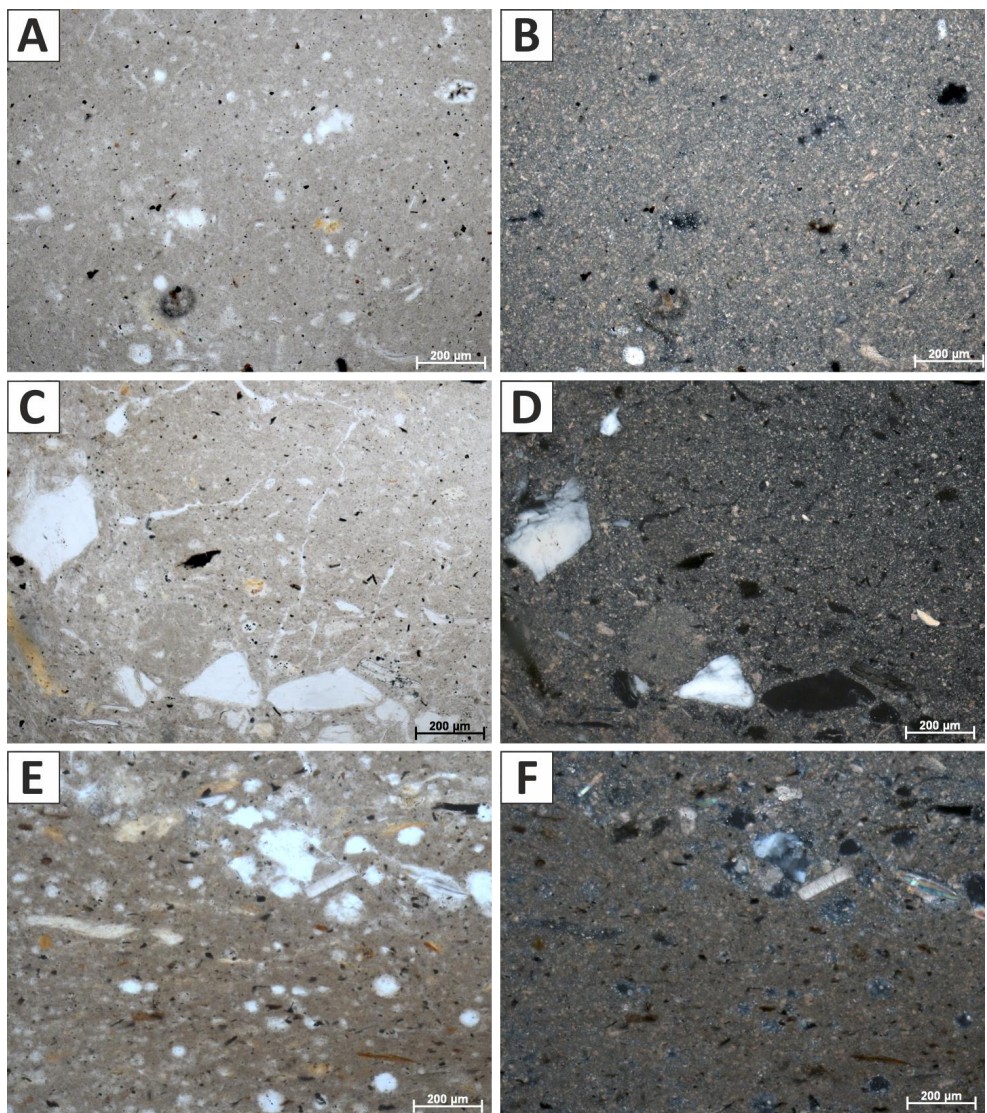

**Figure 2: (A) Microfabric of Unit I at 5 cm depth in transmitted light. Note the randomly oriented, opaque and brownish plant particles. The microcrystalline matrix is more apparent in polarized light (B). (C) Microfabric overview of Unit II at 17 cm depth. Large, up to fine sand-scale detrital feldspar grains occur in layers. (D) Same image section in polarized light. (E) Microfabric of Unit III at 28 cm, illustrating the rather compacted shape of the matrix and the elongated appearance of plant detritus. The layering is evident by the occurrence of larger detrital grains in the upper image part. (F) Same section in polarized light.**

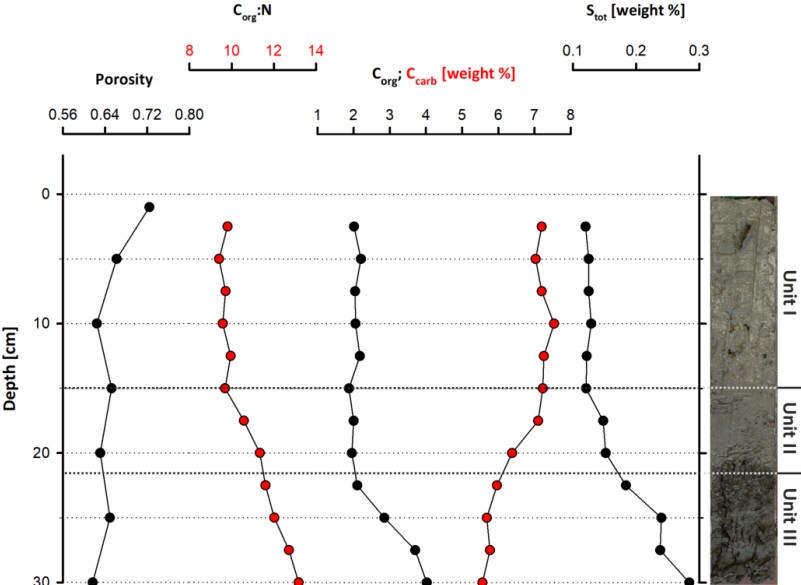


**Figure 3: Geochemical parameters through Core LN-K04, showing an increasing amount of organic carbon, total sulfur**


**and a decreasing porosity with depth.**


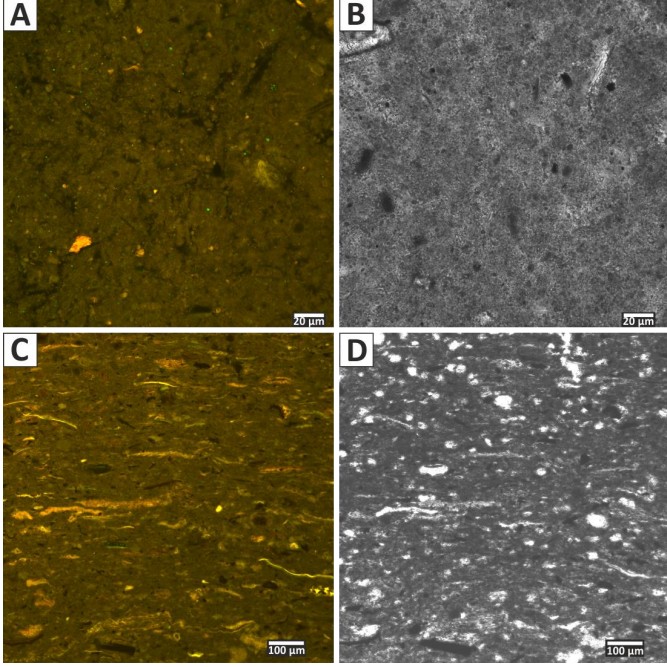


**Figure 4: (A) Laser scanning micrograph (excitation 365 nm/emission 397-700 nm) of Unit I microfabric at 2 cm depth.**


**The small, and randomly orientated plant particles show bright fluorescence due to their chlorophyll content. (B) Same**


**section in transmitted light. (C) Fluorescent texture of Unit III (at 28 cm depth) is visible. The higher amount of plant**


**detritus, particle layering and a compacted matrix are notable. Voids are resin embedding artefacts. (D) Same section**


**as in (C) under transmitted light.**


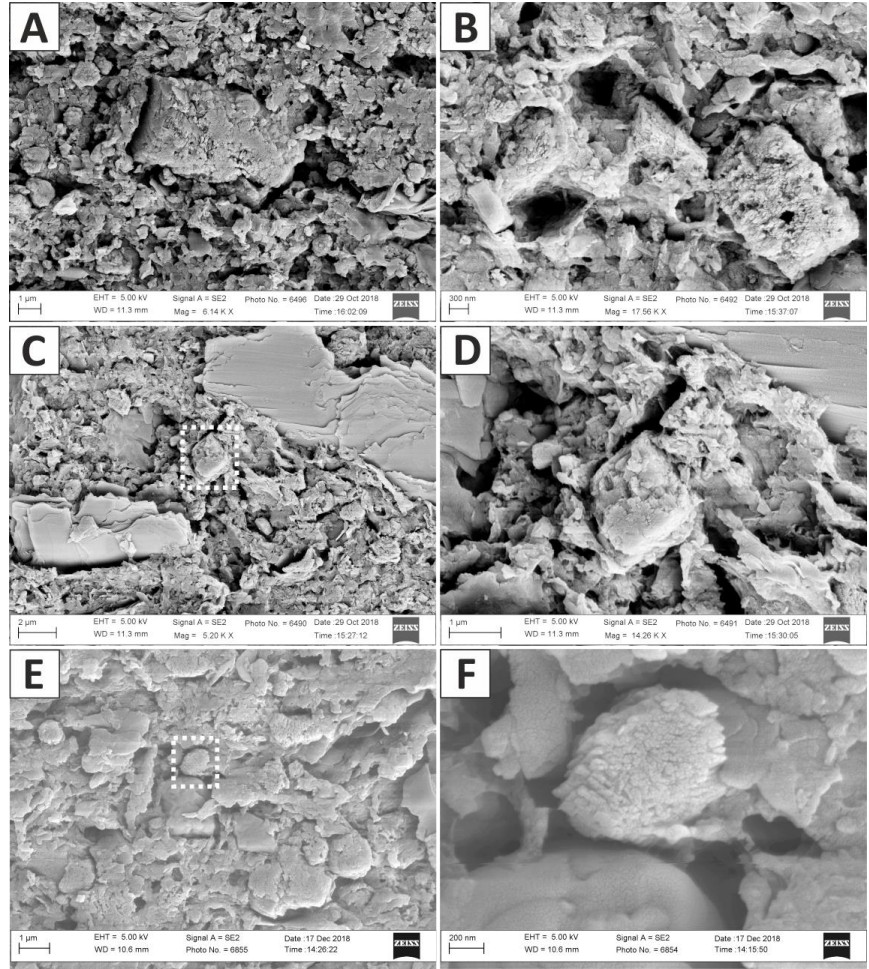


**Figure 5: SEM images of Core LN-K 05, showing the crystal morphology of Ca-Mg phases with increasing depth. (A)**
**HMC/VHMC/dolomite crystal in 9 cm depth. (B) Aggregate of 3 HMC/VHMC/dolomite crystals in 17 cm depth. (C)**
**Matrix overview containing microcrystalline crumbs, layered mica crystals and a HMC/VHMC rhombohedron**
**(indicated by dashed rectangle) at 17 cm depth. (D) Detail of rhombohedron visible in C. (E) Matrix overview in 27 cm**
**depth. HMC/VHMC carbonate crystals appear rather xenomorphic (indicated by dashed rectangle). (F) Close up of**
**HMC/VHMC crystal accentuated in (E).**

331





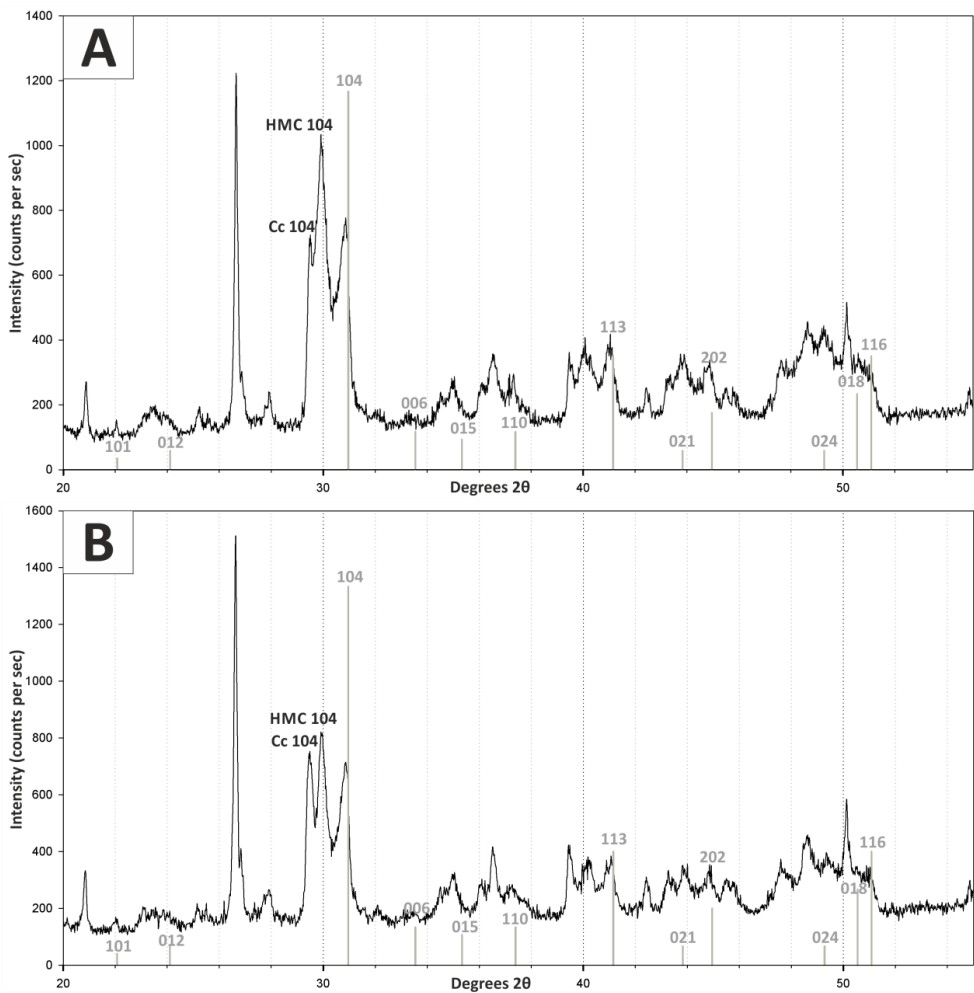

Figure 6: X-ray diffractograms of bulk Lake Neusiedl sediment (A) from 2 cm and (B) from 27.5 cm depth. Positions of dolomite ordering peaks are marked in grey. Position of major calcite (Cc 104) and high-magnesium-calcite (HMC 104) peaks are also indicated. Note that no ordered dolomite can be identified in the investigated Lake Neusiedl samples.

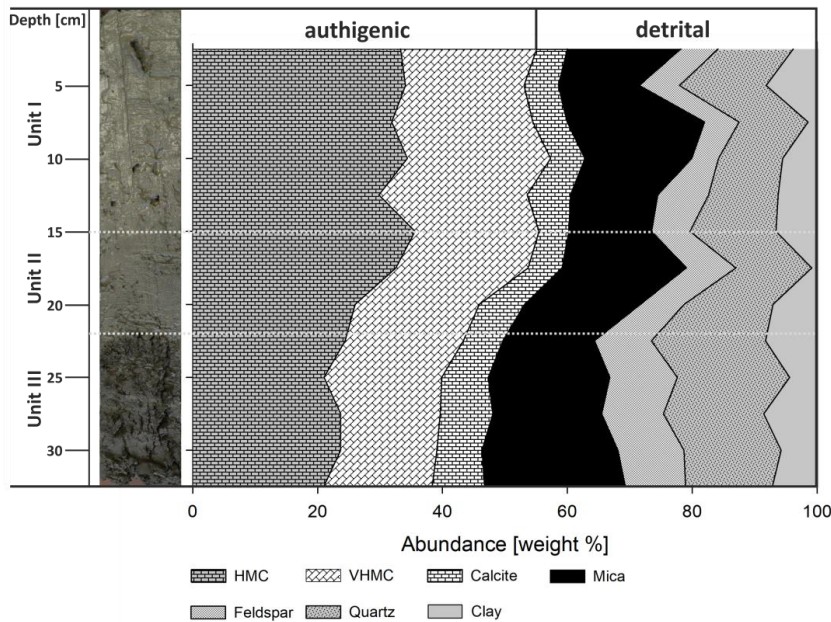

**Figure 7: Core LN-K04 with the defined units I-III (left) and mineral quantities estimated from main peak heights (right; HMC: high-magnesium calcite, VHMC: very-high-magnesium calcite). The changes of mineral abundances coincide with unit boundaries.**

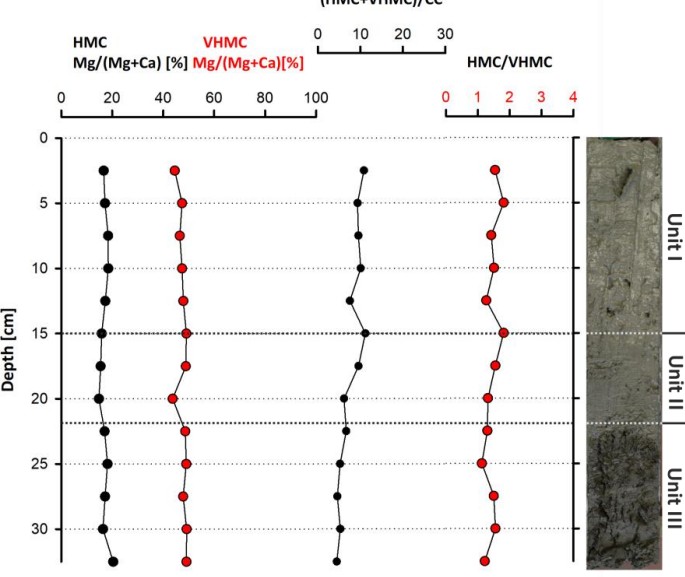

**Figure 8: Stoichiometric compositions of authigenic carbonate phases (HMC and VHMC), their abundance ratio, and their relation to detrital calcite.**





### 4.2 Pore water chemistry

The water chemistry of Lake Neusiedl is characterized by high pH values (9.02) and moderate salinity (1.8‰). Sodium ($Na^+$) and magnesium ($Mg^{2+}$) are the major cations with concentrations of 14.3 and 5.1 mmol·$L^{-1}$, respectively. Calcium ($Ca^{2+}$) concentration is considerably lower at 0.3 mmol·$L^{-1}$. Total alkalinity (TA) measures 11.2 meq·$L^{-1}$ whereas other major anions like chloride ($Cl^-$) and sulfate ($SO_4^{2-}$) hold a concentration of 7 and 4 mmol·$L^{-1}$, respectively. Nutrient ($NH_4^+$, $NO_2^-$, $PO_4^{3-}$, $\Sigma H_2S$, $SiO_{2(aq)}$) concentrations lie below 0.004 mmol·$L^{-1}$.

The pore water chemistry strongly differs between the sediment and the water column. The pH drops significantly at the water-sediment interface to a value around 7.5, which stays constant throughout the sediment core (Fig. 9A). The entire section is anoxic with a redox potential of -234 mV at the top, which increases to -121 mV at the bottom (Fig. 9B). $Na^+$ and $Cl^-$ contents continuously increase with depth from 14 to 20 and from 7 to 8.8 mmol·$L^{-1}$, respectively (Fig. 9A). $Mg^{2+}$ and $Ca^{2+}$ show a different pattern: From 5 to 10 cm depth, the $Mg^{2+}$ content decreases from 5 to 4 mmol·$L^{-1}$, whereas the $Ca^{2+}$ content increases from 0.5 to 0.6 mmol·$L^{-1}$ in the same increment. From 10 cm downwards, the $Mg^{2+}$ content scatters around 4 mmol·$L^{-1}$ and the $Ca^{2+}$ content decreases from 0.6 to below 0.5 mmol·$L^{-1}$ (Fig. 9A). Dissolved $SO_4^{2-}$ and hydrogen sulfide ($\Sigma H_2S$) also show a noticeable trend: The $\Sigma H_2S$ content is close to zero in the top 5 cm of the sediment column, rapidly increases to 1 mmol·$L^{-1}$ between 5 and 10 cm b.s. and remains constant to the bottom of the section. $SO_4^{2-}$ follows an opposite trend. Its concentration decreases from 4 to 1 mmol·$L^{-1}$ in the upper 10 cm b.s. and remains constant at 1 mmol·$L^{-1}$ towards the section bottom. Total alkalinity also increases towards the lower part of the section, from 11.2 to 16.8 meq·$L^{-1}$, with an increase between 5 and 15 cm depth.

$NO_2^-$ is present in the upper 10 cm of the core and reaches its highest value (0.9 µmol·$L^{-1}$) at 2 cm b.s., while its concentration decreases to zero below 10 cm b.s.. Dissolved iron ($Fe^{2+}$) has a similar trend in the upper 10 cm b.s., reaching its highest concentration at a depth of 2 cm (1.4 µmol $L^{-1}$). Below 10 cm core depth, iron concentrations lie below 0.3 µmol $L^{-1}$, with the exception of an outlier value of 0.5 µmol $L^{-1}$ at 13 cm b.s.. Concentrations of ammonia ($NH_4^+$) and phosphate ($PO_4^{2-}$) increase with depth. In the uppermost part of the sediment column, they are close to zero and increase to 0.37 and 0.02 mmol·$L^{-1}$ at 13 cm. These values remain constant to the bottom of the core. Dissolved silica shows a curved profile with 0.3 mmol·$L^{-1}$ at the top, reaching a maximum at 15 cm depth with 0.8 mmol·$L^{-1}$ and declines to concentrations around 0.5 mmol·$L^{-1}$. Methane ($CH_4$) concentration also shows a curved trend, reaching its highest value of 227 µmol·$L^{-1}$ at a depth of 20 cm and concentrations between 14 and 64 µmol·$L^{-1}$ close to the sediment surface (5 and 1 cm, respectively). Dissolved inorganic carbon (DIC) increases from 11.71 mmol·$L^{-1}$ at the top to 18.01 mmol·$L^{-1}$ at 30 cm depth. Only in the 15 to 20 cm increment, the amount of DIC slightly decreases from 15.37 to 14.94 mmol·$L^{-1}$.

According to PHREEQC calculations, the water column at the sampling site (Bay of Rust) is supersaturated with respect to aragonite (SI = 0.92), calcite (SI = 1.07), VHMC (SI = 2.92; protodolomite) and dolomite (SI = 3.46; Fig. 10). Sediment pore water is close to equilibrium throughout the whole section with respect to aragonite, whereas calcite is in equilibrium to slightly supersaturated between 10 and 27.5 cm depth. VHMC (dolomite d) reaches equilibrium between 2.5 and 5 cm, while dolomite is supersaturated in the entire section. It should be noted that all saturation graphs reveal parallel trends, with their highest saturation at 17.5 cm and their lowest at 2.5 cm depth.

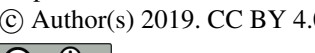



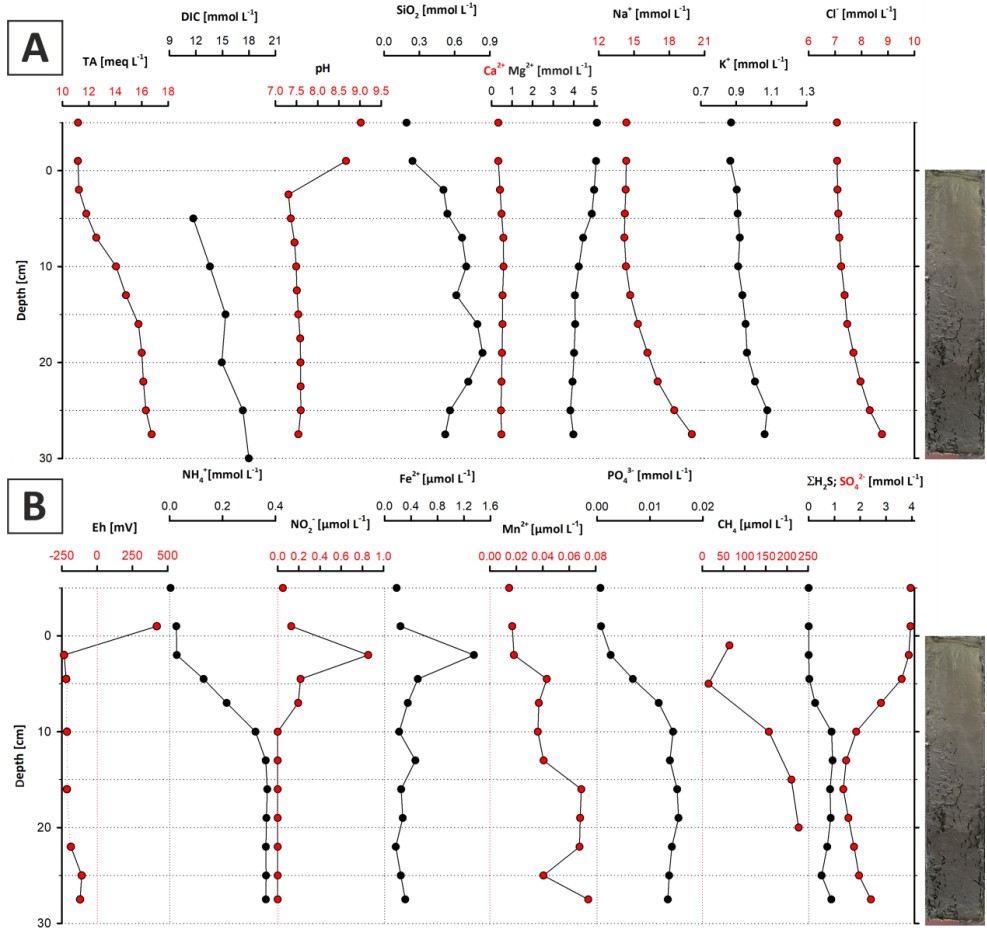

**Figure 9: Major ion- (A) and metabolite concentrations (B) in the pore water of core LN-K03. Note that the sample slightly above 0 cm depth represents the supernatant water, and the top data points represent the water column (see text for explanations).**

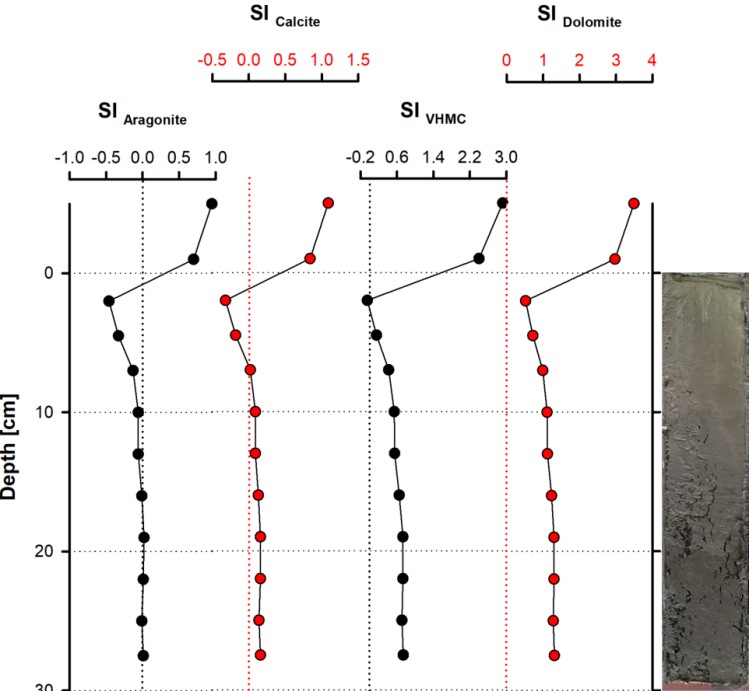

**Figure 10: Saturation indices (SI) of selected carbonate mineral phases. Noteworthy, all phases are clearly supersaturated in the water column but close to saturation throughout most of the sediment column (except for the uppermost 10 cm).**

### 4.3 Bacterial community composition

Bacterial 16S rRNA gene analysis revealed the presence of a diverse bacterial community with 1,226 amplicon sequence variants (ASVs) clustered at 100% sequence identity within the water column, 2,085 to 2,467 ASVs in the top 20 cm of the sediment core and 1,417 to 1,581 ASVs in the deeper sediment (20 - 35 cm core depth). The different bacterial taxa were grouped by known metabolic properties of characterized relatives, listed in Whitman (2015) and additional literature (see supplementary data). The distribution of the most abundant bacterial taxa differs between the water column and the sediment (Fig. 11A and B).

The water column is dominated by aerobic heterotrophs, mainly *Alphaproteobacteria* and *Actinobacteria*, which are only of minor abundance in the sediment. Among the *Alphaproteobacteria*, the SAR 11 clade capable of oxidizing C1-compounds (Sun et al. 2011), is predominant. The nitrogen-fixing *Frankiales* are the most abundant representatives of the *Actinobacteria*. Furthermore, coccoid *Cyanobacteria (Synechococcales)* and *Bacteroidetes* are present in high relative abundances in the water column.

Within sediment Unit I (0-15 cm b. s.), the bacterial community composition changes to mainly anaerobic and facultatively anaerobic taxa. Only the uppermost 5 cm show increased relative abundances of *Cyanobacteria (Synechococcales)* and *Bacteroidetes* (aerobes and facultative anaerobes; Alderkamp et al., 2006, Flombaum et al., 2013), as well as *Verrucomicrobia* (mostly aerobic and facultative anaerobic heterotrophs, He et al. 2017), which include nitrogen-fixing members (Chiang et al. 2018). Besides these groups, *Gammaproteobacteria*,



408 *Acidobacteria*, *Chloroflexi*, as well as sulfate-reducing *Deltaproteobacteria* are abundant. The latter mainly consist

409 of *Desulfobacteraceae* and *Desulfarculales* (Fig. 11C and D).

410 In sediment Unit II (15-22 cm b. s.), the relative proportions of these groups show a transition between sediment

411 unit I and III. While *Gammaproteobacteria*, *Acidobacteria* and *Deltaproteobacteria* are still abundant, the relative

412 abundance of *Chloroflexi* increases strongly from 24.29 to 35.43%. Within the SRB, *Desulfobacteraceae* and

413 *Desulfarculales* are successively replaced by *Deltaproteobacteria* of the Sva0485 clade. The *Syntrophobacterales*

414 show their maximum relative abundance within sediment unit II.

415 In sediment unit III (22-40 cm b. s.), the abundance of *Chloroflexi* further increases to form the dominant bacterial

416 phylum. The phylum is consists of *Dehalococcoidia* and *Anaerolineae*. Other abundant groups in this unit are

417 *Acidobacteria*, *Gammaproteobacteria*, and *Deltaproteobacteria* of the Sva0485 clade. Further details of the

418 microbial community composition are given in von Hoyningen-Huene et al. (2019).





419

**Figure 11: Most abundant taxa in Core 01 (A) and Core 02 (B). The legend indicates all abundant taxa on the phylum level, including the class level for Proteobacteria and Firmicutes. All orders below 0.5% relative abundance were summarized as rare taxa. The abundant taxa change at the transitions from water column to sediment and the lithological units (I-III). The taxonomic composition of sulfate reducers in Core 01 (C) and Core 02 (D) changes gradually from Unit I to II and more pronouncedly from Unit II to III. Sulfate reducers are shown on the class and order level. The column thickness relates to the sampled increments of either 5 or 2.5 cm. Sulfate reducers represent up to 15% of the total bacterial community and were normalized to 100% relative abundance to illustrate the changes within their composition**

428





**5. Discussion**

**5.1 Pore water gradients and their effect on Ca-Mg carbonate supersaturation**

Concentrations of the conservative trending ions $Na^+$, $K^+$, and $Cl^-$, steadily increase towards the bottom of the core section, reaching 19, 1, and 9 mmol·$L^{-1}$, respectively. These concentrations are considerably higher than in the water column, where these ions measure 14, 0.9 and 7 mmol·$L^{-1}$. Moreover, $SO_4^{2-}$ shows an increase near the bottom of the core and is reported to further increase to values of 6.5 mmol·$L^{-1}$ in a longer section from a different locality in the Bay of Rust (not shown in this study), which is higher than the overlying lake water (3.9 mmol·$L^{-1}$). This rise in ion concentration indicates an ion source below the sampled interval. While saline deep ground waters are known to be present in deep aquifers (Neuhuber, 1971; Blohm, 1974; Wolfram, 2006), it is also possible that more highly concentrated brine exists in deeper mud layers due to more recent evaporation events (Fig. 12). Lake Neusiedl dried out entirely between 1865 and 1875 (Moser, 1866) and high ion concentrations may relate to thin evaporite layers and brines that formed during this event.

The cause of the exceptionally high Mg:Ca ratio, which reaches values around 15 in the water column, is not yet entirely understood. The low $Ca^{2+}$ concentrations in Lake Neusiedl can be linked to calcium carbonate formation (e.g. Wolfram and Herzig, 2013), but the for oligohaline, soda-type lakes uncommonly high amounts of $Mg^{2+}$ ions and their source remain elusive. Boros et al. (2014) describe similar phenomena in small alkaline lakes of the western Carpathian plain and relate the high magnesium levels to local hydrogeological conditions and the geological substrate of the lakes.

It should be noted that the Mg:Ca ratio reaches values around 7 in the 5-10 cm increment of the pore water section. This is caused by a considerable decrease of the $Mg^{2+}$ ions in this increment (from 5 to 4 mmol·$L^{-1}$) and an increase in $Ca^{2+}$ concentration (from 0.3 to 0.5 mmol·$L^{-1}$). This effect can be partly explained by a transition zone between lake and pore water in this section, in which the concentration gradient is balanced. Other factors contributing to this concentration shift may include ion exchange, e.g. with $NH_4^+$ generated in the pore water at clay minerals (von Breymann et al., 1990; Celik et al., 2001). However, in the case of Lake Neusiedl, the $NH_4^+$ concentration is not sufficient to explain this change within the Mg:Ca ratio. Another factor causing the decrease of $Mg^{2+}$ concentrations may be the supply of dissolved silica for the precipitation of clay mineral precursor phases (Birsoy, 2002). Increasing $SiO_2$ concentration with depth indicates the dissolution of diatom frustules, which have been observed in thin sections of the present study. It is not entirely clear if this $SiO_2$ release into the pore water is related to hydrochemical or biogenic parameters. As the $SiO_2$ increase in the upper 20 cm of the pore water neither clearly correlates with alkalinity, nor with the salinity gradients (concentrations of conservative ions) and pH is not predictive (Ryves et al., 2006), diatom dissolution by an evident chemical undersaturation (saturation indices of amorphous $SiO_2$ lie between -1.35 and -0.65) may be not the only driver for the $SiO_2$ release. It is also conceivable that the enhanced silica release in the pore water is caused by bacteria, which attack the organic matrix of diatom frustules and, thus, expose the silica bearing skeletons to chemical undersaturation (Bidle and Azam, 1999). Bidle et al. (2003) have linked enhanced dissolution potential to uncultured *Gammaproteobacteria*. This phylum showed increased abundances in the upper sediment column, supporting the hypothesis of a biogenic contribution to diatom dissolution and, hence, the provision of $SiO_2$ to sequester $Mg^{2+}$ (Fig. 12, eq. (5)) in Lake Neusiedl´s pore waters.





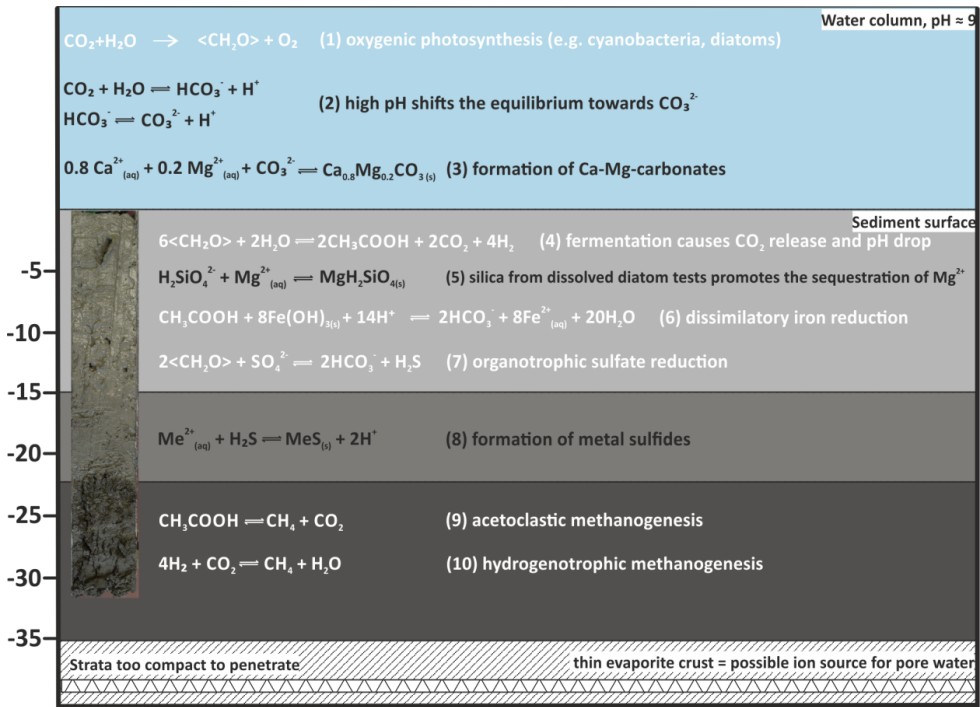

**Figure 12: Suggested major microbial (simplified, indicated in white) and geochemical processes in water- and sediment column of Lake Neusiedl.**

### 5.2 Microbial activity and carbonate saturation

Microbial metabolic reactions strongly affect pore water chemistry, particularly pH, alkalinity and hence carbonate mineral saturation state. In the present approach, the assessment of bacterial community composition is based on the metagenomic DNA within the sediment. This contains the active bacterial communities at their current depth as well as deposited, dormant or dead cells that originated in the water column or at shallower sediment depth (More et al., 2019). In the present study, a background of dormant or dead cells is evident through ASVs belonging to strict aerobes (e.g. *Rhizobiales*, *Gaiellales*) that were detected within deeper parts of the anaerobic mud core (Fig. 11; Suppl. Material Tab. S5).

The water column is characterized by aerobic heterotrophs, including C1-oxidizers (SAR11 clade of the *Alphaproteobacteria*) and highly abundant freshwater *Actinobacteria*. These are common in most freshwater environments. An impact on carbonate mineral saturation or nucleation, however, is unknown as their role in the biogeochemical cycles remains largely undescribed (Neuenschwander, et al., 2018). A high abundance of *Cyanobacteria* of the *Synechococcales* is present in the water column. *Synechococcales* are known to create favorable conditions for carbonate nucleation in alkaline environments by raising the pH, photosynthetic metabolism and the complexation of cations at their cell envelopes (Thompson and Ferris, 1990). Further research is required to verify their potential role in HMC or VHMC formation in Lake Neusiedl.

In sediment unit I (0-15 cm b. s.) *Synechococcales,* as well as aerobic *Bacteroidetes* are still abundant in the top 5 cm, likely due to the sedimentation of their cells from the water column. The uppermost measurement at 2.5 cm depth revealed reducing conditions and a low, close to neutral pH. This supports heterotrophic metabolisms and





fermentation by *Gammaproteobacteria, Acidobacteria, Chloroflexi,* and *Deltaproteobacteria,* which are the major
taxa at this depth. At the very top of the sediment, a peak in $NO_2^-$ and $Fe^{2+}$ points to nitrate-reduction and $Fe^{3+}$-
reduction (Kotlar et al., 1996; Jørgensen and Kasten, 2006). Farther below, the successive increase in $NH_4^+$ and
$PO_4^{3-}$ reflects anaerobic bacterial decomposition of organics, consistent e.g. with *Chloroflexi* capable of
dissimilatory nitrate reduction to ammonium (DNRA).
Sulfate-reducers are present in unit I. Their increasing relative abundance coincides with a decrease in $SO_4^{2-}$ and
an increase in $\Sigma H_2S$ (Fig. 9). Despite a concomitant increase in alkalinity, the bulk metabolic effect of the microbial
community keeps the pH and carbonate saturation low (Fig. 12, eq (7)). Model calculations in aquatic sediments
have shown that sulfate reduction initially lowers the pH (e.g. Soetart et al., 2007) and as the alkalinity increases,
the pH converges at values between 6 and 7. As a consequence, the saturation index for carbonate minerals
concomitantly drops. If a sufficient amount of sulfate is reduced (>10 mmol·$L^{-1}$), the saturation level recovers and
may slightly surpass initial conditions (Meister, 2013). Only when sulfate reduction is coupled to anaerobic
oxidation of methane (AOM), the effect of both would raise the pH to higher values. However, as methane occurs
below 10 cm (Fig. 10), where $SO_4^{2-}$ is still present, AOM is incomplete or absent.
In sediment unit II (15-22 cm b. s.) and unit III (22-40 cm b. s.), the bacterial community composition shifts
towards a high abundance of *Chloroflexi* (*Dehalococcoidia* and *Anaerolineae*), known for their involvement in
carbon cycling as organohalide respirers and hydrocarbon degraders (Hug et al., 2013). This change may reflect
an increase in poorly degradable organic electron donors and hence plant debris in the laminated core unit III. The
change in the relative composition of different orders within the SRB (i.e., change from *Desulfobacterales* and
*Desulfarculales* to Sva0485 and *Spirochaetales*) may also be related to a change in available organic substrates.
In total, sulfate reduction remains high, also recognizable by the occurrence of opaque (sulfide-) mineral spots and
the increase of $S_{tot}$ in the lower part of the section (Fig. 2E; Fig. 3). Fermentation as well as sulfate-reduction
remain high with increasing depth, indicated by the near-neutral pH and raised alkalinity at low carbonate mineral
saturation.

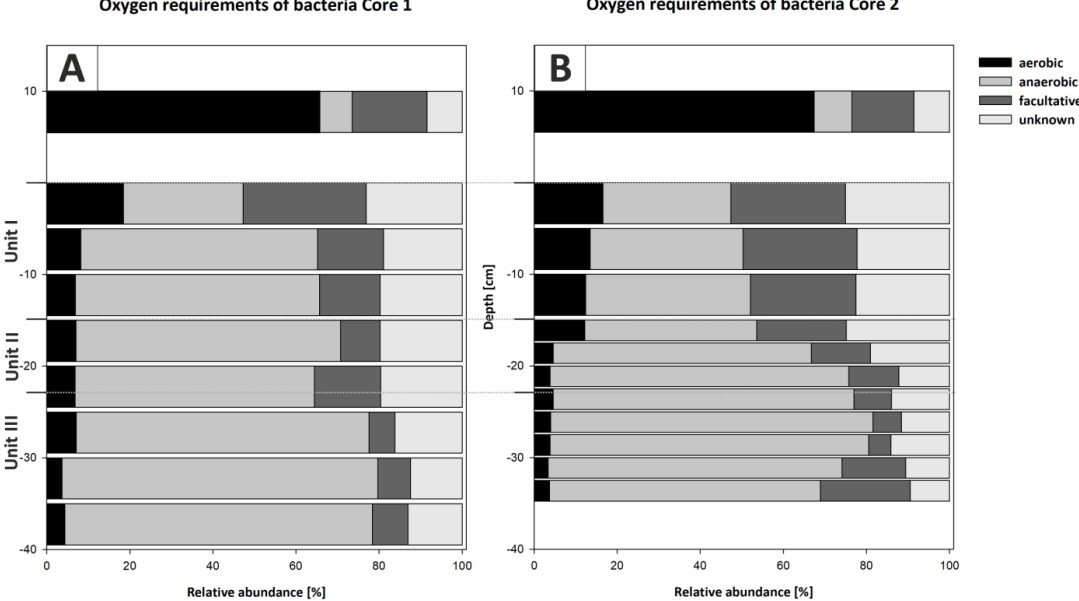


**Figure 13: Oxygen utilization within the most abundant members of the bacterial community (A) and the potential**
**energy metabolisms (B) plotted versus depth in Cores 01 and 02. The community in the water column indicates a**
**predominantly aerobic regime. Rare taxa (< 0.5% relative abundance) were removed from the analysis and abundances**
**normalized to 100%. Bacteria with an unknown metabolism were grouped as unknowns. The community inhabiting**
**the sediment shows an early onset of sulfate reduction in the upper sediment layers and a shift to fermentation at the**
**transition from Unit II to Unit III.**
**5.3 Time and depth of carbonate formation**
A significant difference in saturation state between the water column and the sediment is evident. Whilst the water
column is supersaturated with respect to aragonite, HMC, VHMC and dolomite, they are close to equilibrium in
the pore water. The down shift of saturation from the water column to the pore water is to be expected, due to the
onset of anaerobic, heterotrophic metabolic activity (Fig. 12, eq (4)).
The absence of aragonite at Lake Neusiedl is not entirely clear, as its formation is commonly linked to an interplay
between high temperature, mineral supersaturation and Mg:Ca ratios (Fernández-Díaz et al., 1996; Given and
Wilkinson, 1985). Based on precipitation experiments by De Choudens-Sanchez and Gonzalez (2009), which
include temperatures of 19.98 °C and Mg:Ca ratios up to 5, aragonite would be the favored phase in Lake Neusiedl,
as the lake's Mg:Ca ratio of 15 is too high and the concomitant calcite saturation not sufficient to provide calcite
growth. However, the mentioned experiments were performed in a precipitation chamber with degassing
conditions and hence reduced $\rho CO_2$, which makes them incomparable to the present study. In contrast, Niedermayr
et al. (2013) observed the preferential formation of calcite at high Mg:Ca ratios when an amino acid (polyaspartic
acid) is present. As the water column bears numerous bacterial species (Fig. 11) and potentially comparable
organic compounds, this is a likely scenario for Lake Neusiedl. Nevertheless, the precise evaluation why aragonite,
is not present is impossible, as no related analytical data from the water column are available.
According to Löffler (1979), magnesium calcite forms first, which then alters to protodolomite/VHMC. The
alteration takes place from the inside, hence, resulting in a VHMC core and a HMC rim. However, the observation

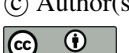



that ratios of HMC to VHMC remain constant around 40 to 50% indicates no significant diagenetic alteration in
the uppermost 30 cm of the sediment. Abrupt changes in these ratios, along with changing contributions of detrital
mineral phases, such as mica and quartz, rather suggest changing sedimentation. Likewise, (low-Mg-) calcite
essentially depends on the input of ostracod shells and transport of detrital carbonates delivered from the catchment
area. Furthermore, no significant diagenetic overprint in form of recrystallization and/or cementation is apparent
from the applied light- and electron-optical methods as well as the geochemical gradients. Most importantly, the
stoichiometric ratio of each carbonate phase remains constant, confirming that no large-scale recrystallization of
these phases occurs.
Considering that no signs of carbonate precipitation or diagenetic alteration were observed in the sediment column
from the Bay of Rust, it can be concluded that carbonate minerals are unlikely to form in the pore water. Instead
Ca-Mg carbonate crystals may precipitate in the water column and are deposited at the bottom of the lake (Fig. 12,
eq (3)). Age estimations for the mud sediments range from 150 years (Löffler, 1979) to 850-2300 years before
present (radiocarbon ages from Neuhuber et al., 2015). Our dataset indicates that authigenic Ca-Mg Carbonate
does not necessarily form in its present location, which is consistent with the large discrepancy between sediment-
and authigenic carbonate age.
The observed detrital mineral spectrum reflects the mineral composition of the adjacent Leitha- (mica, feldspar,
quartz, calcite) and Rust Hills (calcite) and are either windblown or transported by small, eastbound tributaries
(Löffler, 1979). The layering in the lower part of the section (Unit III) reflects the lack of homogenization by wind
driven wave action and indicates a higher water level. As this unit also contains higher amounts of plant particles
and siliciclastics, possibly due to a higher water influx from vegetated surroundings, it is conceivable that the
deposition of Unit III reflects environmental conditions before the installation of the water level regulating Einser-
Kanal in 1909. The increase of $C_{org}$ with depth further reflects this depositional change. It fits the increasing amount
of plant particles with depth. The lignin bearing plant particles are difficult to degrade for heterotrophic organisms
under the prevailing anoxic conditions (Benner et al., 1984). The higher amounts of plant material may reflect a
lower salinity and thus higher primary production at their time of deposition, which can also be related to the
stronger water level oscillations before regulations, including a larger lake surface and almost a magnitude higher
catchment area (refer to a map in the supplementary data, provided by Hegedüs (1783)). Based on this
consideration one might concur with the sediment age estimation of circa 150 years, as proposed by Löffler (1979).
Nevertheless, it is important to distinguish between actual mineral formation and sediment deposition, including
relocation: An unpublished sediment thickness map (GeNeSee project; unpubl.) suggests a current-driven
relocation of mud deposits in the south-western lake area, where the bay of Rust is located. Thus, the radiocarbon
data from Neuhuber et al. (2015) possibly reflect the date of precipitation, whereas Löffler´s age estimation may
refer to the date of local mud deposition.

## 5.4 Potential pathways of authigenic Ca-Mg carbonate formation

The precise formation pathway of authigenic Ca-Mg carbonate mineral precipitation in Lake Neusiedl has been
controversially discussed. Some authors suggest a precipitation of HMC in the water column and subsequent
alteration to VHMC or dolomite within the anoxic pore water of the sediment (Müller et al., 1972). Others suggest
the direct formation of VHMC in the water column (Schiemer and Weisser, 1972). Our XRD and geochemical
data support the latter hypothesis, as no diagenetic alteration is retraceable throughout the sediment section. While
low saturation or even undersaturation in the sediment precludes a microbially induced precipitation in the pore



water, high supersaturation in the surface water body would support precipitation in the water column. Given the
high alkalinity, $CO_2$ uptake by primary producers may have contributed to the high pH and high supersaturation
in the surface water.
An alternative explanation to the controversially discussed microbial dolomite formation would be the ripening
under fluctuating pH conditions in the water column. Deelman (1999) has demonstrated in his precipitation
experiments that dolomite forms if the pH varies. At times of strong supersaturation, metastable carbonates
(protodolomite) are formed, which ripen to ordered dolomite during subsequent phases of undersaturation of the
metastable carbonate (while the stable phase remains supersaturated). This observation reflects Ostwald's step
rule, according to which the metastable phase always forms first. Ostwald's step rule can also be demonstrated in
the pore water, which is buffered by the metastable phase. Thereby the formation of the stable phase (dolomite) is
inhibited despite its supersaturation. This observation is comparable with Land's (1998) "failure" to form dolomite
for 30 years despite 1000-fold supersaturation.
In Lake Neusiedl, fluctuation of the pH in the overlying water column is likely to occur due to variations in
meteoric water input and temperature, which may cause episodes of undersaturation. A fact, which is supported
by Wolfram and Herzig (2013), who report an increase of $Ca^{2+}$ concentration, depending on a dissolution of Ca-
carbonates in Lake Neusiedl's open water during the winter months, when water levels rise and temperatures
decrease. Such a seasonal dependent formation mechanism has recently been suggested to explain dolomite
formation in a Triassic evaporative tidal flat setting (Meister and Frisia, 2019). Alternatively, Moreira et al. (2004)
proposed that undersaturation of metastable phases occurs as a result of sulfide oxidation near the sediment surface.
While we traced only small abundances of sulfate-oxidizing bacteria near the sediment-water interface (1%),
fluctuating hydro-chemical conditions are likely to occur in the diffusive boundary layer, where a pH drop is
observed as a result of the biogeochemical processes discussed above. Dolomite formation in the diffusive
boundary layer has been observed in Lake Van (McCormack et al., 2018), and was interpreted as a result of
abundant microbial EPS, linked to a changing water level and hence -chemistry. In Lake Neusiedl, the amount of
EPS in the diffusive boundary layer is difficult to estimate, but the potential Ca-Mg carbonate favoring change in
hydrochemistry is granted.
**6. Conclusions**
Two phases of Ca-Mg carbonates (HMC, VHMC) as well as calcite occur in form of fine-grained mud in Lake
Neusiedl. Bacterial metabolic activity, including sulfate reduction and fermentation, leads to a decrease of pH
within the sediment, leaving the Ca-Mg-carbonate phases at low/minor saturation in the pore water. In contrast,
Ca-Mg carbonate phases are highly supersaturated in the alkaline water column. There, the carbonate formation
mechanism may involve fluctuating hydrochemical conditions, leading to periods of undersaturation and ripening
of HMC to VHMC. Further, carbonate precipitation may be supported by phototrophic uptake of $CO_2$ by
cyanobacteria, e.g by *Synechococcus*. Precipitation of Ca-Mg carbonate, thus, most likely occurs in the open water.
Based on the presented data set, precipitation or diagenetic alteration within the sediment is not indicated. The
precise Ca-Mg carbonate reaction pathway needs further evaluation.



**Data availability**

All data required for the presented plots and supplementary, analytical data were submitted to PANGEA (Data Publisher for Earth & Environmental Science, doi to be assigned). Microbiological datasets can be requested from Avril von Hoyningen-Huene.

**Author contributions**

Dario Fussmann, Patrick Meister and Andreas Reimer investigated, formally analyzed and curated the hydro- and geochemical data. Avril Jean Elisabeth von Hoyningen-Huene investigated the bacterial communities, and formally analysed and curated the data together with Dominik Schneider and Dario Fussmann. Hana Babková, Andreas Maier and Robert Peticzka conducted data curation. Dario Fussmann wrote the original draft, which was reviewed and edited by Patrick Meister, Avril Jean Elisabeth von Hoyningen-Huene, Andreas Reimer, Dominik Schneider, Gernot Arp and Rolf Daniel. Gernot Arp and Rolf Daniel conceptualized the study, acquired the funding, administered and supervised the project.

**Competing interests**

The authors declare that they have no conflict of interest.

**Acknowledgements**

We thank Wolfgang Dröse, Birgit Röring, and Axel Hackmann for their support during lab work. Furthermore, we thank Susanne Gier for support during XRD measurements and Beatrix Bethke, Caroline Haberhauer, and Barbara Hofbauer for help during sampling. We also thank Erich Draganits, Regina and Rudolf Krachler, and Stephanie Neuhuber for insightful discussions.

**Financial support**

The project was funded by the German Research Foundation DFG, research unit FOR-1644 "CHARON" (subproject TP7: AR 335/8-1, DA 374/11-1). Further support was provided by the Open Access Publication Funds of the Göttingen University. P.M. received funding by the European Commission (Marie-Curie IEF Project TRIADOL; no. 626025) and by the Department of Geodynamics and Sedimentology at the University of Vienna.



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
