# Peer review of "Authigenic formation of Ca-Mg carbonates in the shallow alkaline Lake Neusiedl, Austria"

_Biogeosciences, 2019_

## Referee Comment (RC1) · Anonymous Referee #1 · 13 Dec 2019

In this work, Fussmann and couathors studied the formation of Ca-Mg carbonates in a shallow lake in Austria combining a classical geological approach and microbiology studies. They used a wide vary of methods to study the lake sediments, lake water and pore water: optical microscopy, SEM, EDX, XRD, ion chromatography and ICP-MS. Furthermore, they also studied the bacterial communities on lake water and sediments (but this is beyond my field of expertise). They obtained interesting results on the Ca-Mg carbonate formation of Lake Neusiedl. They proved that Ca-Mg carbonate minerals are formed on the water column and not as post-sedimentary process. After minor corrections this work merits publication in Biogeosciences. Below, I have listed some specific comments that might be helpful for such revision.

General comments: Authors should increase evidence to claim that dolomite is not

present on lake sediments. The evidence provided is not clear enough to distinguish between VHMC and dolomite in the diffractograms shown (see comments below). I am not convinced that the precipitate is VHMC and not dolomite (even when I expect sediments to be VHMC and not dolomite). Failure to demonstrate that precipitates are not dolomite would require rewriting part of the discussion.

Specific comments: Page 9, lines 294-306: What was the criterion to determine that the precipitate was VHMC and not dolomite? Authors claim that due to the shift of 10.4 peak of ordered dolomite, lake sediments are VHMC. However, non-stoichiometric ordered dolomites also occur in nature, showing a shift of 10.4 for lower 2thetha values (if Ca mole > 50%) or for higher 2thetha values (if Ca mole < 50%).

Page 13, figure 6: Looking at this figure, where authors marked the position of dolomite "ordering peaks", one might think that samples have dolomite. As can be seen in both diffractograms, dolomite ordering peaks (i.e., 10.1 and 01.5) seem to be present, indicating that order dolomite can be found on the lake sediments. Could those peaks belong to other phases? A complete list of identified diffraction peaks could be provided in the supplementary material to demonstrate that such peaks do not belong to dolomite. In figure caption should be indicated that such list can be found in supplementary material.

Page 13, caption figure 6: Please change "Positions of dolomite ordering peaks..." for "Position of ordered dolomite peaks..." or "Position of dolomite peaks...". Ordering peaks are the superstructure peaks, i.e., those that are present in dolomite diffractograms but not in calcite diffractograms. Such peaks are reflections with h0.l and 0k.l, with odd-numbered l (Lippmann, 1973).

Page 25, lines 582-590: What are daily and/or seasonal pH variations of the lake? Deelman (1999) performed experiments with variations of pH from $\sim$6 (CO2 bubbling) to $\sim$8 (degas) and during degasification process solutions were kept at 38$°$C. Changes on these conditions can result in longer times for dolomite precipitation. Interestingly,

recent papers claim that dolomite formation could require several million years (Zohdi et al., 2014; Kell-Duivestein et al.,2019). In other words, dolomite could be found in deeper sediments of Lake Neusiedl. Did authors analyse deeper sediments? If not, I hope authors will continue to investigate this interesting lake in the future.

References: Kell-Duivestein, I. J., Baldermann, A., Mavromatis, V., Dietzel, M. Controls of temperature, alkalinity and calcium carbonate reactant on the evolution of dolomite and magnesite stoichiometry and dolomite cation ordering degree - An experimental approach. Chemical geology (In press).

Zohdi, A., Moallemi, S. A., Moussavi-Harami, R., Mahboudi, A., Richter, D. K., Geske, A., Nickandish, A. A., Immenhauser, A. (2014) Shallow burial dolomitization of an Eocene carbonate platform, southeast Zagros Basin, Iran. GeoArabia, 19(4), 17-54.

---

## Referee Comment (RC2) · Anonymous Referee #2 · 21 Dec 2019

This study reports the results of a comprehensive analysis of the chemistry, mineralogy and microbiology of lakewater and sediments in Neusiedlersee, a rather special lake regarding both its physical features (very shallow for its large surface area, no natural outlet, half of it covered by reed) and geochemistry (rather salty and alkaline, with extremely high Mg/Ca ratios). Since the formation of dolomite (or dolomite-like carbonates) have been reported long ago from the lake, the topic is important for getting a better grip on freshwater dolomite formation.

In addition, for the first time, the authors go beyond studying the minerals and chemistry, and provide a detailed characterization of the microbial communities both in water and sediments. To my knowledge, it is also a first that geochemical and mineralogical data are available at high spatial (depth) resolution from the sediment of this lake. In

this sense, the study is also important for a general understanding of what is happening in the mud of shallow, alkaline lakes.

The study is very carefully designed and executed, and the results and their discussion are fully backed up by the data that are reported meticulously. Some of the results are quite surprising: no aragonite is detected even though it should form under these conditions and, contrary to previous and widely held views that dolomitization takes place in the sediment, the findings suggest that high-magnesian carbonates form in the water body. The results raise the possibility of partial dissolution in the sediment and reprecipitation in the water, when particles are stirred up by winds. The authors also point out the significance of fluctuations in pH and water chemistry as possible drivers of changes in the amount of Mg that can be incorporated into the precipitating carbonates.

In summary, this is an interesting and very carefully executed study that deserves to be published in Bigeosciences, after minor corrections.

I have no substantial concerns regarding the scientific content (with one exception: I agree with Anonymous Referee #1 comments about dolomite ordering reflections and the possible presence of dolomite - since this issue is put forward in detail by the Referee, I do not repeat it here). Some minor, editorial comments are below.

55: put comma after "(Zhang et al., 2013b)"

60: lower case after full stop

71: "interface characterized by varying..."

67-77: I think the statement about breaking Ostwald's step rule needs some explanation.

90: put comma after citation

92: "which addresses" is inconsistent with the subjective of the sentence

118: "315 km2" (exponential missing)

121: "comparison"

132: "km2"

140: delete comma after "noteworthy"

143: "the thickness of soft sediment.."

305: "Notably, neither authigenic Ca-Mg carbonate phase shows any..."

308-313, Figure 2 cation, ocurring 3 times: instead of "polarized light", it should be "under crossed polars" or a similar phrase, since probably the left-hand images were also taken in polarized light (but without the analyzer).

378: I do not understand what "dolomite d" refers to.

443: "but the for.." is strange, please reword the sentence

453-466: This is an interesting discussion about dissolved silica. Is there a chance that clay minerals (such as smectite) can precipitate in situ?

---

## Author Comment (AC1) · 31 Jan 2020

Dear anonymous Referee #1,

we thank you for the clear and thorough review. The authors agree that the displayed XRD Spectra do not provide unequivocal evidence regarding a possible cation ordering of the VHMC phase. However, the supposed 01.5 dolomite ordering peak in figure 6 rather belongs to phyllosilicate phases like muscovite and illite. Furthermore, the 10.1 reflection belongs to Ca-Na feldspar phases. To support this statement, an excel file with XY-Processed XRD data and a figure with peaks of abundant mineral phases is added to the digital supplement folder. Nevertheless, as the sediment-powder spectra include a certain amount of noise, a "non-stoichiometric-dolomite" as defined by Sibley

et al. (1994) cannot be fully excluded in this study.

Page 9, lines 294-306: What was the criterion to determine that the precipitate was VHMC and not dolomite? Authors claim that due to the shift of 10.4 peak of ordered dolomite, lake sediments are VHMC. However, non-stoichiometric ordered dolomites also occur in nature, showing a shift of 10.4 for lower 2Θ values (if Ca mole > 50%) or for higher 2Θ values (if Ca mole < 50%).

Reply: The authors agree that the shift of the 10.4 peak alone is not evidence enough to prove the absence of dolomite. A new figure, added to the digital supplement folder, indicates all identified mineral peaks. In this figure, no superstructural ordering peak of dolomite is observed.

Page 13, figure 6: Looking at this figure, where authors marked the position of dolomite "ordering peaks", one might think that samples have dolomite. As can be seen in both diffractograms, dolomite ordering peaks (i.e., 10.1 and 01.5) seem to be present, indicating that order dolomite can be found on the lake sediments. Could those peaks belong to other phases? A complete list of identified diffraction peaks could be provided in the supplementary material to demonstrate that such peaks do not belong to dolomite. In figure caption should be indicated that such list can be found in supplementary material.

Reply: An excel file with XY processed XRD data and a figure with peaks of abundant mineral phases has been added. It clearly shows that the peaks found at 22 and 35° 2Θ belong to detrital phyllosilicates (muscovite, illite) and Ca-Na feldspar (anorthite), but do not represent the dolomite ordering peaks 10.1 and 01.5.

Page 13, caption figure 6: Please change "Positions of dolomite ordering peaks. . ." for "Position of ordered dolomite peaks. . ." or "Position of dolomite peaks. . .". Ordering peaks are the superstructure peaks, i.e., those that are present in dolomite diffractograms but not in calcite diffractograms. Such peaks are reflections with h0.l and 0k.l, with odd-numbered l (Lippmann, 1973).

Reply: Caption is now changed to "Position of dolomite peaks..."

Page 25, lines 582-590: What are daily and/or seasonal pH variations of the lake? Deelman (1999) performed experiments with variations of pH from âĹij6 ($CO_2$ bubbling) to âĹij8 (degas) and during degasification process solutions were kept at 38âŮęC. Changes on these conditions can result in longer times for dolomite precipitation. Interestingly, recent papers claim that dolomite formation could require several million years (Zohdiet al., 2014; Kell-Duivestein et al., 2019). In other words, dolomite could be found in deeper sediments of Lake Neusiedl. Did authors analyse deeper sediments? If not, I hope authors will continue to investigate this interesting lake in the future

Reply: The statement about fluctuating hydrochemical conditions is based upon observations by Wolfram and Herzig (2013), who processed monitoring data obtained by the "Biologische Station Neusiedler See". These authors noticed temperature and pH changes during the winter months. They provide a mean value of 8.8 for the years 1998-2009 and mention fluctuations between annual pH values of 8.0-9.1. Unfortunately, Wolfram and Herzig did not publish the processed monitoring data set. Nevertheless, the attached table ("figure-2.pdf) provides examples of accessible pH data, which were taken in the open water of Lake Neusiedl. These data e. g. show a pH difference of 0.6 (8.5-9.1) within the year 1959.

On the one hand, the authors did not analyze deeper sediments of Lake Neusiedl, because the unconsolidated, lacustrine mud is placed directly upon coarse, semi-consolidated Pannonian strata (Loisl et al., 2018). The latter substrate was simply too hard to penetrate with the applied coring method. On the other hand, Lake Neusiedl is only of Holocene age and its sedimentary record thus comprises approximately 13000 years (Herzig and Dokulil, 2001). Based on this fact, deeper authigenic sediments and longer precipitation- or maturation times of Ca-Mg-carbonates to dolomite, as mentioned by Zohdi et al. (2014) can be excluded in this study.

References:

Dinka, M., Über die regionalen wasserchemischen Verschiedenheiten des ungarischen Seeteil im Neusiedler See, BFB-Bericht, 79, 31-39, 1993

Dinka, M., Ágoston-Szabó, E., Berczik, Á. and Kutrucz, G.: Influence of water level fluctuation on the spatial dynamic of the water chemistry at lake Ferto/Neusiedler See, Limnologica, 34, 1-2, https://doi.org/10.1016/s0075-9511(04)80021-5, 2004

Herzig, A., and Dokulil, M.: Neusiedler See - ein Steppensee in Europa., in: Ökologie und Schutz von Seen., edited by: Dokulil, M., Hamm, A., and Kohl, J.-G., Facultas-Universitäts-Verlag, Wien, 401, 2001

Loisl, J., Tari, G., Draganits, E., Zámolyi, A. and Gjerazi, I.: High-resolution seismic reflection data acquisition and interpretation, Lake Neusiedl, Austria, northwest Pannonian Basin, Interpretation, 6, 1, https://doi.org/10.1190/int-2017-0086.1, 2018

Schroll, E. and Wieden, P.: Eine rezente Bildung von Dolomit im Schlamm des Neusiedler Sees, Tschermaks mineralogische und petrographische Mitteilungen, 7, 286, https://doi.org/10.1007/bf01127917, 1960

Sibley, D.F., Nordeng, S.H. and Borkowski, M.L.: Dolomitization kinetics of hydrothermal bombs and natural settings. Journal of Sedimentary Research, 64, 3a, https://doi.org/10.1306/d4267e29-2b26-11d7-8648000102c1865d, 1994

Stehlik, A., Österr. Akad. Wiss. Math-Naturwiss. Kl. Sitzungsber., Abt I, 180, 8-10, 1972

Stehlik, A., Biolog. Fosch. Stat. Neusiedersee, BFB-Bericht, 13, 1976

Wolfram, G. and Herzig, A.: Nährstoffbilanz Neusiedler See. Wiener Mitteilungen, 228, 317-338, 2013

Zohdi, A., Moallemi, S. A., Moussavi-Harami, R., Mahboudi, A., Richter, D. K., Geske, A., Nickandish, A. A., Immenhauser, A.: Shallow burial dolomitization of

an Eocene carbonate platform, southeast Zagros Basin, Iran. GeoArabia, 19, 4, https://doi.org/10.1007/s10347-014-0423-3, 2014

Please also note the supplement to this comment:
https://www.biogeosciences-discuss.net/bg-2019-449/bg-2019-449-AC1-supplement.zip

[Figure]

[Figure]

**Fig. 1.**

| pH | Date [month/year] | Data type | Reference |
|---|---|---|---|
| 8.8 | 1958 | annual mean | Schroll and Wieden (1959) |
| 8.8 | 09/1958 | single value | Stehlik (1972) |
| 8.75 | 12/1958 | single value | Stehlik (1972) |
| 8.5 | 02/1959 | single value | Stehlik (1972) |
| 8.97 | 07/1959 | single value | Stehlik (1972) |
| 9.0 | 09/1959 | single value | Stehlik (1972) |
| 8.9 | 10/1959 | single value | Stehlik (1972) |
| 9.1 | 11/1959 | single value | Stehlik (1972) |
| 8.7 | 06/1970 | single value | Stehlik (1972) |
| 8.63 | 12/1971 | single value | Stehlik (1976) |
| 8.44 | 12/1972 | single value | Stehlik (1976) |
| 8.62 | 02/1974 | single value | Stehlik (1976) |
| 8.7 | 11/1974 | single value | Stehlik (1976) |
| 9.0 | 07/1991 | single value | Dinka (1993) |
| 8.9 | 08/1991 | single value | Dinka (1993) |
| 8.5 | 07/1994 | single value | Dinka et al., (2004) |
| 8.1 | 07/1996 | single value | Dinka et al., (2004) |
| 9.5 | 07/2002 | single value | Dinka et al., (2004) |
| 8.7 | 1998-2009 | mean value from 11 annual means | Wolfram and Herzig (2013) |
| 9.02 | 08/2017 | single value | this study |

---

## Author Comment (AC2) · 31 Jan 2020

Dear anonymous Referee #2, we thank you for the clear and thorough review. The authors agree with both referees that the displayed XRD spectra do not provide unequivocal evidence for a possible cation ordering of the VHMC phase. Nevertheless, the supposed-to 01.5 dolomite ordering peak in figure 6 belongs to phyllosilicate phases like muscovite and illite. Furthermore, the 10.1 reflection fits Ca-Na feldspar phases (anorthite). To support this statement, an excel file with XY-Processed XRD data and a figure with peaks of abundant mineral phases is added to the digital supplement folder. As the sediment-powder spectra include a certain amount of noise, a "non-stoichiometric-dolomite" as defined by Sibley et al. (1994) cannot be fully excluded in this study. We further acknowledge the editorial comments. If they are not mentioned

below, they are included, according to your suggestions, in the new manuscript version.

67-77: I think the statement about breaking Ostwald's step rule needs some explanation.

Reply: We agree that the statement of Deelman (1999) about "breaking Ostwald's step rule" needs clarification. In fact, Ostwald's step rule is not really "broken". It is always valid. It is rather a consequence of Ostwald's step rule that under fluctuating conditions the metastable phase forms during high supersaturation and the stable phase (dolomite) forms during low supersaturation by replacement of the metastable phase. Thus, a new sentence will be paraphrased accordingly, e.g. "...dolomite can form due to such fluctuations in pH and temperature, according to Ostwald's step rule via undersaturation of other metastable carbonate phases."

378: I do not understand what "dolomite d" refers to

Reply: In a previous version of figure 10, disordered dolomite (dolomite d) was used instead of VHMC as a caption. The term "dolomite d" is deleted in the present manuscript version.

453-466: This is an interesting discussion about dissolved silica. Is there a chance that clay minerals (such as smectite) can precipitate in situ?

Reply: The possibility of in-situ smectite formation is difficult to evaluate, as no data for dissolved aluminium are available. In any case, the precipitation of Mg-clays, i.e. sepiolite, is favored in the open water of Lake Neusiedl, e.g. due to the high pH and Magnesium content (Galán and Pozo, 2011). PHREEQC calculations provide a SI of 3.4 for sepiolite in the open water. In contrast, lower pH values argue against an authigenic Mg-clay formation in the pore water ($SI_{sepiolite}$ varies between -1 and -2). Furthermore, clay minerals commonly form when amorphous silica is supersaturated (Birsoy, 2002), which is not the case in the investigated sediment cores ($SI_{SiO_2}$ = -0.5; less diatom tests observed in the lower core, $SiO_2$ release into the pore water). All in

all, there is currently no indication of in-situ (authigenic) smectite or Mg-clay formation in the pore waters due to undersaturation and no identification of sepiolite in the XRD-Spectra.

References: Birsoy, R.: Formation of sepiolite-palygorskite and related minerals from solution, Clays and Clay Minerals, 50, 664 736, https://doi.org/10.1346/000986002762090263, 2002.

Deelman, J.: Low-temperature nucleation of magnesite and dolomite, Neues Jahrbuch Fur Mineralogie Monatshefte, 289, 1999.

Galán, E. and Pozo, M.: Palygorskite and Sepiolite Deposits in Continental Environments. Description, Genetic Patterns and Sedimentary Settings, Developments in Clay Science, 3, 6, htpps://doi.org/ 10.1016/B978-0-444-53607-5.00006-2, 2011.

Sibley, D.F., Nordeng, S.H. and Borkowski, M.L.: Dolomitization kinetics of hydrothermal bombs and natural settings. Journal of Sedimentary Research, 64, 3a, https://doi.org/10.1306/d4267e29-2b26-11d7-8648000102c1865d, 1994.

Please also note the supplement to this comment:
https://www.biogeosciences-discuss.net/bg-2019-449/bg-2019-449-AC2-supplement.zip

―――――――――――――――――

[Figure]

[Figure]

**Fig. 1.**

---

## Author Response (AR1)

**Point by Point Response to Reviews**

**Dear anonymous Referee #1**, we thank you for the clear and thorough review.

The authors agree that the displayed XRD Spectra do not provide unequivocal evidence regarding a possible cation ordering of the VHMC phase. However, the supposed 01.5 dolomite ordering peak in figure 6 rather belongs to phyllosilicate phases like muscovite and illite. Furthermore, the 10.1 reflection belongs to Ca-Na feldspar phases. To support this statement, an excel file with XY-Processed XRD data and a figure with peaks of abundant mineral phases is added to the digital supplement folder. Nevertheless, as the sediment-powder spectra include a certain amount of noise, a "non-stoichiometric-dolomite" as defined by Sibley et al. (1994) cannot be fully excluded in this study.

**Page 9, lines 294-306:** What was the criterion to determine that the precipitate was VHMC and not dolomite? Authors claim that due to the shift of 10.4 peak of ordered dolomite, lake sediments are VHMC. However, non-stoichiometric ordered dolomites also occur in nature, showing a shift of 10.4 for lower $2\Theta$ values (if Ca mole > 50%) or for higher $2\Theta$ values (if Ca mole < 50%).

**Reply:** The authors agree that the shift of the 10.4 peak alone is not evidence enough to prove the absence of dolomite. A new figure, added to the digital supplement folder, indicates all identified mineral peaks. In this figure, no superstructural ordering peak of dolomite is observed.

**Page 13, figure 6:** Looking at this figure, where authors marked the position of dolomite "ordering peaks", one might think that samples have dolomite. As can be seen in both diffractograms, dolomite ordering peaks (i.e., 10.1 and 01.5) seem to be present, indicating that order dolomite can be found on the lake sediments. Could those peaks belong to other phases? A complete list of identified diffraction peaks could be provided in the supplementary material to demonstrate that such peaks do not belong to dolomite. In figure caption should be indicated that such list can be found in supplementary material.

**Reply:** An excel file with XY processed XRD data and a figure with peaks of abundant mineral phases has been added. It clearly shows that the peaks found at 22 and 35° $2\Theta$ belong to detrital phyllosilicates (muscovite, illite) and Ca-Na feldspar (anorthite), but do not represent the dolomite ordering peaks 10.1 and 01.5.

**Page 13, caption figure 6:** Please change "Positions of dolomite ordering peaks..." for "Position of ordered dolomite peaks..." or "Position of dolomite peaks...". Ordering peaks are the superstructure peaks, i.e., those that are present in dolomite diffractograms but not in calcite diffractograms. Such peaks are reflections with h0.l and 0k.l, with odd-numbered l (Lippmann, 1973).

**Reply:** Caption is now changed to "Position of dolomite peaks..."

**Page 25, lines 582-590:** What are daily and/or seasonal pH variations of the lake? Deelman (1999) performed experiments with variations of pH from ~6 ($CO_2$ bubbling) to ~8 (degas) and during degasification process solutions were kept at 38°C. Changes on these conditions can result in longer times for dolomite precipitation. Interestingly, recent papers claim that dolomite formation could require several million years (Zohdiet al., 2014; Kell-Duivestein et al., 2019). In other words, dolomite could be found in deeper sediments of Lake Neusiedl. Did authors analyse deeper sediments? If not, I hope authors will continue to investigate this interesting lake in the future

**Reply:** The statement about fluctuating hydrochemical conditions is based upon observations by Wolfram and Herzig (2013), who processed monitoring data obtained by the "Biologische Station Neusiedler See". These authors noticed temperature and pH changes during the winter months. They provide a mean value of 8.8 for the years 1998-2009 and mention fluctuations between annual pH values of 8.0-9.1. Unfortunately, Wolfram and Herzig did not publish the processed monitoring data set. Nevertheless, the table below provides examples of accessible pH data, which were taken in the open water of Lake Neusiedl. These data e. g. show a pH difference of 0.6 (8.5-9.1) within the year 1959

| pH | Date [month/year] | Data type | Reference |
|---|---|---|---|
| 8.8 | 1958 | annual mean | Schroll and Wieden (1959) |
| 8.8 | 09/1958 | single measurement | Stehlik (1972) |
| 8.75 | 12/1958 | single measurement | Stehlik (1972) |
| 8.5 | 02/1959 | single measurement | Stehlik (1972) |
| 8.97 | 07/1959 | single measurement | Stehlik (1972) |
| 9.0 | 09/1959 | single measurement | Stehlik (1972) |
| 8.9 | 10/1959 | single measurement | Stehlik (1972) |
| 9.1 | 11/1959 | single measurement | Stehlik (1972) |
| 8.7 | 06/1970 | single measurement | Stehlik (1972) |
| 8.63 | 12/1971 | single measurement | Stehlik (1976) |
| 8.44 | 12/1972 | single measurement | Stehlik (1976) |
| 8.62 | 02/1974 | single measurement | Stehlik (1976) |
| 8.7 | 11/1974 | single measurement | Stehlik (1976) |
| 9.0 | 07/1991 | single measurement | Dinka (1993) |
| 8.9 | 08/1991 | single measurement | Dinka (1993) |
| 8.5 | 07/1994 | single measurement | Dinka et al., (2004) |
| 8.1 | 07/1996 | single measurement | Dinka et al., (2004) |
| 9.5 | 07/2002 | single measurement | Dinka et al., (2004) |
| 8.8 | 1998-2009 | mean value from 11 annual means | Wolfram and Herzig (2013) |
| 9.02 | 08/2017 | single measurement | this study |

On the one hand, the authors did not analyze deeper sediments of Lake Neusiedl, because the unconsolidated, lacustrine mud is placed directly upon coarse, semi-consolidated Pannonian strata (Loisl et al., 2018). The latter substrate was simply too hard to penetrate with the applied coring method. On the other hand, Lake Neusiedl is only of Holocene age and its sedimentary record thus comprises approximately 13000 years (Herzig and Dokulil, 2001). Based on this fact, deeper authigenic sediments and longer precipitation- or maturation times of Ca-Mg-carbonates to dolomite, as mentioned by Zohdi et al. (2014) can be excluded in this study.

**Reply:** We agree that the statement of Deelman (1999) about "breaking Ostwald's step rule" needs clarification. In fact, Ostwald's step rule is not really "broken". It is always valid. It is rather a consequence of Ostwald's step rule that under fluctuating conditions the metastable phase forms during high supersaturation and the stable phase (dolomite) forms during low supersaturation by replacement of the metastable phase.

Thus, a new sentence will be paraphrased accordingly, e.g. "…Precipitation-experiments conducted by Deelman (1999) have shown that dolomite can form due to such fluctuations in pH and temperature. Hence, they agree with Ostwald's step rule, because dolomite formation happens via undersaturation of other metastable carbonate phases.."

**378:** I do not understand what "dolomite d" refers to

**Reply:** In a previous version of figure 10, disordered dolomite (dolomite d) was used instead of protodolomite as a caption. The term "dolomite d" is deleted in the present manuscript version.

**453-466:** This is an interesting discussion about dissolved silica. Is there a chance that clay minerals (such as smectite) can precipitate in situ?

**Reply:** The possibility of *in-situ* smectite formation is difficult to evaluate, as no data for dissolved aluminium are available. In any case, the precipitation of Mg-clays, i.e. sepiolite, is favored in the open water of Lake Neusiedl, e.g. due to the high pH and Magnesium content (Galán and Pozo, 2011). PHREEQC calculations provide a SI of 3.4 for sepiolite in the open water. In contrast, lower pH values argue against an authigenic Mg-clay formation in the pore water ($SI_{sepiolite}$ varies between -1 and -2). Furthermore, clay minerals commonly form when amorphous silica is supersaturated (Birsoy, 2002), which is not the case in the investigated sediment cores ($SI_{SiO2}$ = -0.5; less diatom tests observed in the lower core, $SiO_2$ release into the pore water). All in all, there is currently no indication of *in-situ* (authigenic) smectite or Mg-clay formation in the pore waters due to undersaturation and no identification of sepiolite in the XRD-Spectra.

$CO_2 + H_2O \rightleftharpoons HCO_3^- + H^+$
$HCO_3^- \rightleftharpoons CO_3^{2-} + H^+$    (2) high pH shifts the equilibrium towards $CO_3^{2-}$

$0.8\ Ca^{2+}_{(aq)} + 0.2\ Mg^{2+}_{(aq)} + CO_3^{2-} \rightleftharpoons Ca_{0.8}Mg_{0.2}CO_{3\ (s)}$    (3) formation of Ca-Mg-carbonates

Sediment surface
$6<CH_2O> + 2H_2O \rightleftharpoons 2CH_3COOH + 2CO_2 + 4H_2$    (4) fermentation causes $CO_2$ release and pH drop
$H_2SiO_4^{2-} + Mg^{2+}_{(aq)} \rightleftharpoons MgH_2SiO_{4(s)}$    (5) silica from dissolved diatom tests promotes the sequestration of $Mg^{2+}$
$CH_3COOH + 8Fe(OH)_{3(s)} + 14H^+ \rightleftharpoons 2HCO_3^- + 8Fe^{2+}_{(aq)} + 20H_2O$    (6) dissimilatory iron reduction
$2<CH_2O> + SO_4^{2-} \rightleftharpoons 2HCO_3^- + H_2S$    (7) organotrophic sulfate reduction

$Me^{2+}_{(aq)} + H_2S \rightleftharpoons MeS_{(s)} + 2H^+$    (8) formation of metal sulfides

$CH_3COOH \rightleftharpoons CH_4 + CO_2$    (9) acetoclastic methanogenesis
$4H_2 + CO_2 \rightleftharpoons CH_4 + H_2O$    (10) hydrogenotrophic methanogenesis

Strata too compact to penetrate    thin evaporite crust = possible ion source for pore water

[revised manuscript text omitted]